# Comparison of scattering ratio profiles retrieved from ALADIN/Aeolus and CALIOP/CALIPSO observations and preliminary estimates of cloud fraction profiles

Artem G. Feofilov[1], Hélène Chepfer[1], Vincent Noël[2], Rodrigo Guzman[1], Cyprien Gindre[1], Po-Lun Ma[3], and Marjolaine Chiriaco[4]

[1] LMD/IPSL, Sorbonne Université, UPMC Univ Paris 06, CNRS, École Polytechnique, Paris, France
[2] Laboratoire d'Aérologie, CNRS/UPS, Observatoire Midi-Pyrénées, Toulouse, France
[3] Atmospheric Sciences and Global Change Division, Pacific Northwest National Laboratory, Richland, Washington, USA
[4] LATMOS/IPSL, Université Versailles Saint-Quentin en Yvelines, Guyancourt, France

*Correspondence to*: Artem G. Feofilov (artem.feofilov@lmd.polytechnique.fr)

## Abstract

The space-borne active sounders have been contributing invaluable vertically resolved information of atmospheric optical properties since the launch of Cloud-Aerosol Lidar and Infrared Pathfinder Satellite Observation (CALIPSO) in 2006. To
build long-term records from space-borne lidars useful for climate studies, one has to understand the differences between successive space lidars operating at different wavelengths, flying on different orbits, and using different viewing geometries, receiving paths, and detectors. In this article, we compare the results of Atmospheric Laser Doppler INstrument (ALADIN) and Cloud-Aerosol Lidar with Orthogonal Polarization (CALIOP) lidars for the period from 28/06/2019 to 31/12/2019. First, we build a dataset of ALADIN/CALIOP collocated profiles (Δdist < 1º; Δtime < 6h). Then we convert the ALADIN's
355nm particulate backscatter and extinction profiles into the scattering ratio vertical profiles SR(z) at 532 nm using molecular density profiles from Goddard Earth Observing System Data Assimilation System, version 5 (GEOS-5 DAS). And finally, we build the CALIOP and ALADIN globally gridded cloud fraction profiles CF(z) in applying the same cloud detection threshold to the SR(z) profiles of both lidars at the same spatial resolution.

Before comparing the SR(z) and CF(z) profiles retrieved from the two analyzed lidar missions, we performed a numerical
experiment to estimate the best achievable cloud detection agreement $CDA_{norm}(z)$ considering the differences between the instruments. We define $CDA_{norm}(z)$ in each latitude/altitude bin as the occurrence frequency of cloud layers detected by both lidars, divided by a cloud fraction value for the same latitude/altitude bin. We simulated the SR(z) and CF(z) profiles that would be observed by these two lidars if they were flying over the same atmosphere predicted by a global model. By analyzing these simulations, we show that the theoretical limit for $CDA_{norm}^{theor}(z)$ for a combination of ALADIN and CALIOP
instruments is equal to 0.81±0.07 at all altitudes. In other words, 19% of the clouds cannot be detected simultaneously by two instruments due to said differences.

The analyses of the actual observed CALIOP/ALADIN collocated data set containing ~78000 pairs of nighttime SR(z) profiles revealed: (a) the values of SR(z) agree well up to ~3km height; (b) the CF(z) profiles show agreement below ~3km where ~80% of the clouds detected by CALIOP are detected by ALADIN as expected from the numerical experiment; (c) above this height, the $CDA_{norm}^{obs}(z)$ reduces to ~50%; (d) on average, better sensitivity to lower clouds skews the ALADIN's cloud peak height in pairs of ALADIN/CALIOP profiles by ~0.5±0.6 km downwards, but this effect does not alter the heights of polar stratospheric clouds and high tropical clouds thanks to their strong backscatter signals; (e) the temporal evolution of the observed $CDA_{norm}^{obs}(z)$ does not reveal any statistically significant change during the considered period. This indicates that the instrument-related issues in ALADIN L0/L1 have been mitigated at least down to the uncertainties of the following $CDA_{norm}^{obs}(z)$ values: 68±12%, 55±14%, 34±14%, 39±13%, and 42±14% estimated at 0.75 km, 2.25 km, 6.75 km, 8.75 km, and 10.25 km, respectively.

## 1 Introduction

Clouds play an important role in the energy budget of our planet: optically thick clouds reflect the incoming solar radiation, leading to cooling of the Earth, while thinner clouds act as "greenhouse films", preventing escape of the Earth's long-wave radiation to space. Climate feedback analyses show that clouds are a large source of uncertainty for the climate sensitivity of climate models and, so, for the future climate evolution (e.g. Nam et al., 2012; Chepfer et al., 2014; Guzman et al., 2017; Vaillant de Guélis et al., 2018; Zelinka et al., 2020). Understanding the Earth's energy budget requires knowing cloud's coverage, geographical and vertical distribution, temperature, and optical properties.

Satellite observations have been providing a continuous survey of clouds over the whole globe. Infrared sounders have been observing our planet since 1979: from the TIROS Operational Vertical Sounder (TOVS) instruments (Smith, 1979) onboard the National Oceanic and Atmospheric Administration (NOAA) polar satellites to the Atmospheric InfraRed Sounder (AIRS) spectrometer (Chahine et al., 2006) onboard Aqua (since 2002) and to the Infrared Atmospheric Sounder Interferometer (IASI) instrument (Chalon et al., 2001; Hilton et al., 2012) onboard MetOp (since 2006), with increasing spectral resolution. Despite an excellent daily coverage and daytime/nighttime observation capability (Menzel et al., 2016; Stubenrauch et al., 2017), the height uncertainty of the cloud products retrieved from the observations performed by these space-borne instruments is large (e.g. Feofilov and Stubenrauch, 2017). This precludes the retrieval of the cloud's vertical profile with the accuracy needed for climate relevant processes and feedback analysis. This drawback does not exist for active sounders, which measure the altitude-resolved profiles of backscattered radiation with accuracy on the order of 1E0-1E2 meters. Among them, one can name the Cloud-Aerosol Lidar with Orthogonal Polarization (CALIOP) lidar (Winker et al., 2003, 2004, 2007, 2009) and CloudSat radar (Stephens et al., 2002; 2009), which have been providing vertically resolved cloud and aerosol properties since 2006. The CATS (Cloud-Aerosol Transport System) lidar on-board ISS provided measurements for over 33 months starting from the beginning of 2015 (McGill et al., 2015). The Atmospheric Laser Doppler INstrument (ALADIN) lidar on-board Aeolus (Krawczyk et al., 1995; Stoffelen et al., 2005; ADM-Aeolus Science report,

2008) has been measuring horizontal winds and aerosols/clouds since September 2018. More lidars are planned – in 2023, the ATmosperic LIDar (ATLID)/EarthCare instrument (Héliere et al., 2012) will be launched and other space-borne lidars are in the development phase. All active instruments share the same measuring principle – the emitter sends a brief pulse of laser or radar electromagnetic radiation to the atmosphere, and the receiver registers a time-resolved backscatter signal collected through its telescope. However, the wavelength, pulse energy, pulse repetition frequency (PRF), telescope diameter, orbit, detector, and many other parameters are not the same for any pair of instruments. These differences define the active instruments' capability of detecting atmospheric aerosols and/or clouds for a given atmospheric scenario and observation conditions (day, night, averaging distance). At the same time, there is an obvious need to ensure the continuity of global space-borne measurements and to get a smooth transition between the satellite missions (Winker et al. 2017; Chepfer et al., 2014, 2018).

This work seeks to address this issue using ALADIN/Aeolus space-borne wind lidar operating at 355 nm and CALIOP/CALIPSO atmospheric lidar operating at 532 nm. Even though the primary goal of ALADIN is wind detection (Reitebuch et al., 2020; Straume et al., 2020), its products include profiles of atmospheric optical properties. In addition, the methods developed in this study and its conclusions will set the stage for the future comparison of the ATLID/EarthCare observations with other space-borne lidars.

The structure of the article is as follows. In Section 2, we describe the datasets used in this study and explain the collocation criteria. Section 3 provides the definitions and the basic formulae needed for comparison of two lidars operating at different wavelengths. In Section 4, we describe the numerical experiment aimed at the estimation of the best possible theoretically achievable cloud detection agreement for the cloud fraction profiles retrieved from CALIOP and ALADIN observations. Section 5 is dedicated to the analysis of the results and to the discussion of similarities and differences between the collocated SR profiles and cloud fraction distributions. Section 6 concludes the article.

## 2 Data

We start this section with the description of ALADIN/Aeolus optical properties dataset, followed by the description of CALIOP/CALIPSO product. In the next steps, we define the procedures and criteria for the comparison of these two products.

### 2.1 AEOLUS

In this work, we provide only a brief description of the lidar and the details necessary for understanding the key differences between the compared instruments. For a detailed description of the Aeolus mission and its instrument, we refer the reader to (Krawczyk et al., 1995; Stoffelen et al., 2005; ADM-Aeolus Science report, 2008; Flamant et al., 2017). The Aeolus satellite carries a Doppler wind lidar called ALADIN, which operates at 355 nm wavelength and is composed of a transmitter, a Cassegrain telescope, and a receiver capable of separating the molecular (Rayleigh) and particular (Mie) backscattered

photons (high spectral resolution lidar, HSRL). The lidar observes the atmosphere at 35° from nadir and perpendicular to the satellite track, its orbit is inclined at 96.97°, and the instrument overpasses the equator at 6h and 18h of local solar time (LST), see also Fig. 1 and Table 1 to compare with CALIOP.

The laser emitter of the lidar sends 15 ns long pulses of 355 nm radiation down to the atmosphere 50 times per second. The lidar optical system collects the backscattered photons, which are then registered in the instrument's Rayleigh and Mie channels. Wind is detected with the help of interferometric technique from the image formed on the Accumulation Charge Coupled Device (ACCD) detectors of the lidar (Chanin et al., 1989). Besides the winds, the Aeolus processing algorithms retrieve the optical properties of the observed atmospheric layers (Ansmann et al., 2007; Flamant et al., 2017). The vertical resolution of the instrument is adjustable, but the total number of points in a vertical profile is equal to 24, that corresponds to a number of rows in ACCD. The observation priorities changed throughout the period of the mission (ESA, 2021), and for most of the period considered in this work (see below), the vertical sampling of both Mie and Rayleigh channels between 2 km and 22 km was equal to 1 km whereas the sampling below 2 km varied from 0.25 to 1 km. The native horizontal resolution of 140 m of the instrument is sacrificed to achieve higher signal-to-noise ratio (SNR) both onboard by accumulating the detected profiles and on the ground by averaging the downloaded profiles at different steps of the processing chain (Flamant et al., 2017). In Section 4.2, we address the effects of averaging on the accuracy of cloud detection for this instrument and give general recommendations for the future missions.

ALADIN is a relatively new instrument, and its calibration/validation activity is still on the way (Baars et al., 2020; Donovan et al., 2020; Kanitz et al., 2020; Reitebuch et al., 2020; Straume et al., 2020). This includes, but is not limited to internal calibration and comparisons with other observations. The Aeolus mission faced several technical issues, which hindered getting the planned specifications. These issues are related to several factors: (a) laser power degradation (60 mJ/pulse instead of 80 mJ/pulse) and signal losses in the emission and reception paths (33%) that results in lower SNR than planned, (b) telescope mirror temperature effects biasing the wind detection and calibration of Mie and Rayleigh channels of ALADIN, (c) constantly increasing number of hot pixels of both ACCD detectors (Weiler et al., 2021) leading to errors both in wind speed and in retrieved optical parameters of the atmosphere. The Aeolus teams mitigated at least some of these adverse effects (e.g. Baars ql., 2020; Weiler et al., 2021), and it would be interesting to see whether the pilot L2A dataset, Prototype_v3.10 is free of cloud detection quality trends.

We have performed the present study using the pilot L2A dataset from Aeolus, Prototype_v3.10, which is available for participating Cal/Val teams for a limited period of ALADIN's observations, from 28/06/2019 through the 31/12/2019. According to (Flamant et al., 2008, 2017), the L2A data is retrieved from the L1B product of this instrument and it contains height profiles of Mie and Rayleigh co-polarized backscatter and extinction coefficients, scattering ratios (SR), and lidar ratios along the lidar line-of-sight (Flamant et al., 2017; Lolli et al., 2013). For each vertical profile corresponding to a slant path in Fig. 1, we extracted the SR, backscatter, and extinction profiles calculated by standard correct algorithm (Flamant et al., 2017). As for the SR, we draw the reader's attention to the definitions and conversion formulae given below in Section 3.2. The horizontal resolution of this product is 87 km.

The important companions of these profiles are quality flag columns. For our analysis, we kept only the layers, which are

marked either by a high Mie SNR flag or by high Rayleigh SNR flag, and by a flag indicating an absence of signal attenuation. These flags are necessary and sufficient for valid extinction, backscatter, and SR(z) profiles, which we use in the analysis.

## 2.2 CALIPSO-GOCCP

CALIOP, a two-wavelength polarization-sensitive near nadir viewing lidar, provides high-resolution vertical profiles of

aerosols and clouds (Winker et al., 2004, 2007, 2009). Its orbital altitude is 705 km and the orbit is inclined at 98.05°. The lidar overpasses the equator at 1h30 and 13h30 LST, see also Table 1 and the left-hand-side parts of Fig. 1 panels. It uses three receiver channels: one measuring the 1064 nm backscatter intensity and two channels measuring orthogonally polarized components of the 532 nm backscattered signal. Cloud and aerosol layers are detected by comparing the measured 532 nm signal return with the return expected from a molecular atmosphere (see the definitions in Section 3.2).

The General Circulation Model (GCM) Oriented Cloud Calipso Product (CALIPSO-GOCCP) was initially designed to evaluate GCM cloudiness. It is derived from CALIPSO L1/NASA products at Laboratory of Dynamic Meteorology (LMD) and Institute of Pierre-Simon Laplace (IPSL) with the support of NASA/CNES, ICARE Thematic Center (Lille, France), and ClimServ data service (IPSL) and it contains observational cloud diagnostics including the instantaneous scattering ratio (profiles) at the native horizontal resolution of CALIOP (333 m) and at 480m vertical resolution (Chepfer et al., 2008, 2010,

2012). This makes it a good reference dataset for ALADIN retrievals because one can easily recalculate it to ALADIN's horizontal and vertical grids through averaging along the track and in vertical bins, respectively.

## 2.3 Collocation of AEOLUS and CALIPSO profiles

Figure 1 illustrates the orbit and overpass time differences between the two lidars. In Fig. 1b, AEOLUS overflies the same area as was measured by CALIOP ~4.5 h earlier (Fig. 1a). We recall here, that the ALADIN's line of sight is pointed at 35°

to nadir and perpendicular to the flight direction (purple slant paths in the right-hand side parts of Fig. 1 panels) whereas the CALIOP probes the atmosphere in near nadir mode (3° off nadir). As for any collocation, there is a trade-off between the quality of collocation and the number of collocated pairs of profiles. As we show below, for AEOLUS and CALIPSO, one has to supplement this tradeoff with a requirement of a representative geographical coverage, because imposing a stricter temporal overlap criterion adversely affects the latitudinal distribution of the collocated points. Since the horizontal

averaging and resolution of the Aeolus Prototype_v3.10 product is 87 km, there is not much sense in collocating the data with the accuracy better than this value. On the other hand, a fractional standard deviation $f_c$ of cloud water content at 1° (~111 km) distance is about 0.5 for a cloud cover of 1 (Boutle et al., 2014), and there is a risk of comparing incoherent quantities, so we took Δdist = 1° as a limit for the collocations and created several subsets based on the Δtime, the absolute value of the difference between the two collocated measurements. In Fig. 2, we show six such subsets, and the Table 2

provides the total number of collocations for each of them. On the one hand, one can see that a strict collocation criterion of

Δtime < 1h (black curve in Fig. 2) provides the information only about two narrow zones in the Southern and Northern polar regions. On the other hand, an excellent latitudinal coverage corresponding to Δtime < 24 h (dashed magenta curve in Fig. 2) comes at the cost of mixing up the cases, which differ by almost one day that is unacceptable from the point of view of temporal variation. Finally, we have chosen for the analysis a subset corresponding to Δtime < 6h. Over the oceans, the
diurnal effects in cloud distribution associated with this delay are small (e.g. Noël et al., 2018; Chepfer et al., 2019; Feofilov and Stubenrauch, 2019) and the land represents just one third of the analyzed cases. ALADIN observes the atmosphere in dusk-dawn mode whereas CALIPSO has a clear separation between the daytime and the nighttime observations (Fig. 1). To avoid the risks associated with the solar contamination, we picked up only the collocations, which correspond to night-time CALIPSO observations. This yielded about 7.7E4 pairs of SR profiles. In supplementary materials (Feofilov et al., 2021), we
provide the complete collocated database, which corresponds to the last line, 4$^{th}$ column of Table 2 (3.2E5 collocations), for further analysis by the interested teams.

## 3 Method

### 3.1 Lidar equation

An atmospheric lidar sends a brief pulse of laser radiation directed towards the atmosphere. The lidar optics collects the
backscattered photons and drives them to a detector. The detected signal is time-resolved, and each time bin corresponds to a fixed distance from the lidar to the certain atmospheric layer. AEOLUS wavelength of emission is 355nm while CALIPSO emits at 532 nm and 1064 nm. In the atmosphere, the photons coming from the lidar can be backscattered by the molecules, which are much smaller than the wavelength of our two lasers (Rayleigh scattering), by the aerosol particles, which are comparable to or larger than the wavelengths of our two lasers, and by the coarse aerosol and cloud particles, which are
much larger than the wavelengths of our two lasers (Mie scattering). These processes are characterized by the corresponding wavelength-dependent backscatter coefficients $\beta_{mol}(\lambda, z)$ and $\beta_{part}(\lambda, z)$ measured in [m$^{-1}$ sr$^{-1}$]. The attenuation of the laser beam along its path within each layer is characterized by extinction coefficients $\alpha_{mol}(\lambda, z)$ and $\alpha_{part}(\lambda, z)$ in [m$^{-1}$]. On their pathway in the atmosphere, the photons are also scattered in other directions than backscatter and then collected in the telescope after multiple scatterings. The total lidar attenuated backscattered signal (ATB) corrected for geometrical effects
and normalized to molecular signal is usually written as:

$$ATB(\lambda, z) = (\beta_{mol}(\lambda, z) + \beta_{part}(\lambda, z)) \times e^{-2 \int_{Z_{sat}}^{z} (\alpha_{mol}(\lambda, z\prime) + \eta \alpha_{part}(\lambda, z\prime)) dz\prime} \qquad (1)$$

where Z$_{sat}$ is the altitude of the satellite, λ is the wavelength, and η is a multiple scattering coefficient, which depends on the lidar configuration and is set to 0.7 for CALIOP (see (Winker, 2003; Chiriaco et al., 2006; Chepfer et al., 2008; Garnier et al., 2015; Reverdy et al., 2015) for the discussion of this value).

## 3.2 Two definitions of scattering ratio profile

To highlight particles in an atmospheric layer versus molecular background, one often uses the "scattering ratio" or SR. But, two different definitions of SR exist in the literature, and in particular, the ALADIN documents and CALIPSO documents do not use the same definition. So, we provide both definitions and explain our choice below. The first one relates only to scattering properties of the medium and is used in ALADIN product (Flamant et al., 2017):

$$SR^A(\lambda, z) = \frac{\beta_{mol}(\lambda, z) + \beta_{part}(\lambda, z)}{\beta_{mol}(\lambda, z)} \quad (2)$$

According to this definition, $SR^A(\lambda, z)$ is strictly equal to or greater than unity and its interpretation is straightforward: the larger the number, the stronger is the contribution of particles to backscattered signal. But, this definition requires knowledge of both $\beta_{mol}(\lambda, z)$ and $\beta_{part}(\lambda, z)$, which available from ALADIN observations thanks to its HSRL capability (see Section 2.1) but not from non-HSRL lidars such as CALIPSO (see Section 2.2).

The second definition is closer to the profiles observed by classic non-HSRL lidars and is used in CALIPSO products (e.g. Chepfer et al., 2008, 2013):

$$SR^C(\lambda, z) = \frac{ATB(\lambda, z)}{AMB(\lambda, z)} \quad (3)$$

where $ATB(\lambda, z)$ is the total attenuated backscatter given by Eq. 1 and $AMB(\lambda, z)$ is the attenuated molecular backscatter estimated in the absence of particles:

$$AMB(\lambda, z) = \beta_{mol}(\lambda, z) \times e^{-2 \int_{Z_{sat}}^{z} \alpha_{mol}(\lambda, z') \, dz'} \quad (4)$$

The $ATB(\lambda, z)$ values in Eq. 3 are measured by a lidar and the profile of $AMB(\lambda, z)$ can be estimated from Eq. 4 if the molecular density profile is known. Since the exponential part in the enumerator of Eq. 1 leads to a significant attenuation in the presence of particles, the value of $SR(\lambda, z)$ can be less than unity below a thick cloud layer. The definition (Eq. 4) is convenient for the lidars, which cannot distinguish the molecular and particulate components of the backscattered signals. Since the CALIOP is such a lidar, we will use this very definition in the present work as it was done before (Chepfer et al., 2008, 2013). Correspondingly, here and below $SR(\lambda, z) = SR^C(\lambda, z)$. In the rest of the paper, we use the definition of $SR(\lambda, z) = SR^C(\lambda, z)$ given in Eq. 3 and we do not use the $SR^A(\lambda, z)$ given in Eq. 2.

## 3.3 Estimating SR(z) profiles at 532 nm from ALADIN data

After having stated the SR definition (Eq. 3) that we will use to compare the observations collected by the two instruments, we now need to consider their wavelength differences. Indeed, the SR(z) profile is wavelength-dependent as $\alpha_{mol}(\lambda, z)$, $\alpha_{part}(\lambda, z)$, $\beta_{mol}(\lambda, z)$, $\beta_{part}(\lambda, z)$ and, therefore, $ATB(\lambda, z)$, $AMB(\lambda, z)$ depend on the wavelengths. Therefore, one needs to convert the first instrument's SR values to those of the second one, or vice versa. Leaping ahead, we say that since ALADIN can distinguish the molecular backscatter from the particulate one, it provides more information, so it is better suited for the conversion than CALIOP. Below, we provide the formalism used for the conversion (Collis and Russell, 1976;

Bucholz, 1995) as well as the corresponding variable values calculated at two wavelengths, 355 and 532 nm. For the molecular backscatter:

$$\beta_{mol}(\lambda,z) = (d\sigma/d\Omega)_\lambda \times N(z); \ \alpha_{mol}(\lambda,z) = \frac{4\pi}{1.5}\beta_{mol}(\lambda,z) \tag{5}$$

$$(d\sigma/d\Omega)_\lambda = \frac{\sigma(\lambda,z)}{4\pi} \times \frac{3}{4}(1+cos^2(\pi)) \tag{6}$$


$$\sigma(\lambda,z) = \frac{24\pi^3\left(n_s^2(\lambda)-1\right)^2(6+3\rho(\lambda))}{\lambda^4 N_s^2\left(n_s^2(\lambda)+2\right)^2(6-7\rho(\lambda))} \tag{7}$$

where $(d\sigma/d\Omega)_\lambda$ is a differential cross section [m$^2$ sr$^{-1}$], $N(z)$ is a number density [m$^{-3}$], $\sigma(\lambda,z)$ is Rayleigh cross section [m$^2$], $n_s(\lambda)$ is the refractive index for standard air, $\rho(\lambda)$ is the depolarization factor, and $N_s$ is the number density of standard air (2.54743×10$^{25}$ m$^{-3}$). We estimated the $n_s(\lambda)$ values according to (Ciddor, 1996) and obtained $n_s(355nm) = $ 1.00028571 and $n_s(532nm) = 1.00027821$. We took the $\rho(\lambda)$ values from Table 1 of (Bucholz, 1995) according to which

$\rho(355nm) = 3.01 \times 10^{-2}$ and $\rho(532nm) = 2.84 \times 10^{-2}$. The corresponding values of $(d\sigma/d\Omega)_\lambda$ at 355nm and 532nm are then 3.2897988×10$^{-31}$ m$^2$ sr$^{-1}$ and 6.1668318×10$^{-32}$ m$^2$ sr$^{-1}$, respectively. For the particulate backscatter, we took advantage of the fact that the extinction and backscatter coefficients $\alpha_{part}(\lambda,z)$ and $\beta_{part}(\lambda,z)$ barely change at these wavelengths for large particles. Using a known molecular density profile from GEOS-5 DAS (Goddard Earth Observing System Data Assimilation System, version 5), see (Rienecker, 2008), and estimating the $AMB(532nm,z)$ and

$ATB(532nm,z)$ values from Eqs. 1, 2 and Eqs. 4−7, we finally get the $SR'(532nm,z)$ profile for ALADIN, which is comparable with the $SR(532nm,z)$ of the CALIOP:

$$SR'(532nm,z) = \frac{(\beta_{mol}(532,z)+\beta_{part}(355,z))\times e^{-2\int_{Z_{sat}}^{Z}\left(\alpha_{mol}(532,z\prime)+\eta_{355}\alpha_{part}(355,z\prime)\right)dz\prime}}{\beta_{mol}(532,z)\times e^{-2\int_{Z_{sat}}^{Z}\alpha_{mol}(532,z\prime)\,dz\prime}} \tag{8}$$

In Eq. 8, we deliberately kept 355 nm for particle backscatter and extinction coefficients to show their provenance. As for the multiple scattering coefficient $\eta_{355}$, it is usually accepted that the multiple scattering effects can be neglected for

ALADIN with its field of view of just ~0.02 mrad. But, there are indications (e.g. Donovan et al., 2020) that these effects still can affect the observations, so we took it equal to 0.9. We note that the value of $\eta_{355}$ in our conversion (Eq. 8) mostly affects the low-level clouds where, as we will see below, the instruments compare well.

Since the vertical resolution of ALADIN is changing with geographical region and the special range-bin adjustment was performed for certain periods throughout the mission, we recalculated all SR profiles to a regular 1 km vertical grid.

## 3.4. Calculating averaged SR(z) profiles from CALIOP data

Since we used a high-resolution CALIOP data on a 333m horizontal grid, a direct comparison with ALADIN L2 product with its 87 km horizontal averaged data was not possible. To calculate the averaged CALIOP $SR(z)$, we took the original $ATB(z)$ and $AMB(z)$ profiles, averaged them in the ±40 km along the CALIOP orbit track in the vicinity of the point defined as the closest one to ALADIN's track, and got the $SR(z)$ (Eq. 3). The CALIOP vertical binning was also adjusted to

imitate coarse vertical resolution of ALADIN (1 km). These averaging together with the application of a SR cloud detection

threshold may lead to an overestimation of the cloud fraction in the boundary layer, for a field of optically thick geometrically small liquid clouds, e.g. shallow cumulus (see Sect. 4.2) as discussed in Chepfer et al. 2010 and 2013., but this overestimation should be similar for both CALIOP and ALADIN.

In the rest of the article, we discuss the collocated and recalculated SR profiles at 87 km horizontal and 1 km vertical resolution.

### 3.5 Cloud detection, cloud fraction, and normalized cloud detection agreement

In this work, we define an atmospheric layer as cloudy when the following condition fulfills (Chepfer et al., 2013):

$$SR(532nm, z) > 5 \tag{9}$$

For cloud detection, we deliberately do not apply the second criterion of (Chepfer et al., 2013):

$$ATB(532nm, z) - AMB(532nm, z) > 2.5 \times 10^{-6} \, m^{-1} sr^{-1} \tag{10}$$

because of two reasons: (a) this criterion was introduced in (Chepfer et al., 2013) to filter noise in individual profiles at native CALIOP resolution (1/3km along track), whereas in this work we use the $SR(532nm, z)$ averages recalculated from ATB and AMB over ~80 km distance along the track and (b) this would have adversely affected the high cloud amount of ALADIN. Even though this definition makes the CALIOP clouds inconsistent with their definition in current CALIPSO products, this allows estimating the potential capabilities of ALADIN for cloud detection. If a given atmospheric layer was observed multiple times, we define the cloud fraction (CF) in a usual way:

$$CF(z) = \frac{N_{cld}(z)}{N_{tot}(z)} \tag{11}$$

where $N_{cld}(z)$ is a number of times the condition of Eq. 9 fulfills and $N_{tot}(z)$ is a total number of measurements in this layer. As for cloud detection agreement and disagreement, we distinguish four cases: when both CALIOP and ALADIN detect a cloud, when neither of them detects a cloud, when CALIOP detects a cloud whereas ALADIN misses it, and when ALADIN detect a cloud whereas CALIOP misses it. We will name these cases as YES_YES, NO_NO, YES_NO, and NO_YES, and will define their occurrence frequencies as:

$$R_{YES\_YES}(z) = \frac{N_{YES\_YES}(z)}{N_{tot}(z)} ; R_{NO\_NO}(z) = \frac{N_{NO\_NO}(z)}{N_{tot}(z)} ; R_{YES\_NO}(z) = \frac{N_{YES\_NO}(z)}{N_{tot}(z)} ; R_{NO\_YES}(z) = \frac{N_{NO\_YES}(z)}{N_{tot}(z)} \tag{12}$$

The first term in Eq. 12 corresponds to cloud detection agreement ($CDA(z)$), which we will also use in its normalized form, $CDA_{norm}(z)$:

$$CDA_{norm}(z) = \frac{CDA(z)}{CF(z)} = \frac{N_{YES\_YES}(z)}{N_{cld}(z)} \tag{13}$$

As follows from these definitions, if $CF(z)$ is greater than zero and $CDA_{norm}(z)$ is equal to 1 then there is a perfect agreement between the clouds retrieved from both instruments. In the same way, if $CF(z)$ is greater than zero and $CDA_{norm}(z)$ is equal to 0 then there is no agreement.

## 4. Theoretical estimate of the best achievable cloud detection agreement between ALADIN and CALIOP

The aforementioned differences between the missions prevent that the two lidars will observe the same clouds at the same time, except for the polar zones. Knowing the differences in the orbits, wavelengths, and spatial resolution, one can carry out a numerical experiment aimed at the estimation of the best achievable agreement $CDA^{theor}(z)$ and $CDA_{norm}^{theor}(z)$ that one can expect for a combination of these two missions.

### 4.1 Setup of the numerical experiment

To estimate the theoretically possible cloud detection agreement for a considered combination of two lidars and for the chosen collocation criteria, we performed the following numerical experiment outlined in a flowchart in Fig. 3. First, we created a gridded atmosphere from the output of the U.S. Department of Energy's Energy Exascale Earth System Model (E3SM) atmosphere model (EAM) version 1 (EAMv1; Rasch et al., 2019) for the conditions of autumn equinox in Northern hemisphere. This subset does not contain winter atmosphere possible for the period covered by Aeolus Prototype_v3.10 dataset, but it is representative enough from the point of view of the cloud fraction profiles and their variability since it presents a snapshot of both hemispheres, pole-to-pole. From this data, we created a set of daily orbits or "lidar curtains" at the horizontal resolution of CALIOP (333m). Since the resolution of EAMv1 data is coarser than that of CALIOP, we estimated the subgrid cloud variability along the satellite's track using the parameterization of (Boutle et al., 2014) and added it to the data.

Then we fed this high-resolution atmospheric input to the Cloud Feedback Model Intercomparison Project Observational Simulator Package, v2 (COSP2) simulator, which calculates the atmospheric observables for space-borne instruments (Swales et al., 2018). The CALIOP simulator is built into COSP2 (Chepfer et al., 2008) whereas the ALADIN simulator is not yet a part of this package, so we used the 355 nm calculations by COSP2 (initially developed for ATLID Reverdy et al., 2015) at fine grid corresponding to ALADIN's original laser pulse frequency rate (50 Hz). To imitate the diurnal variation, we modulated the SRs using the 6-hour diurnal cycle amplitudes for land and ocean retrieved from active and passive observations (Noël et al., 2018; Chepfer et al., 2019; Feofilov and Stubenrauch, 2019). With these two high-resolution simulations in hand, we created simulated pairs of "collocated" data with the Δdist distribution modulated by that of a real collocated dataset. Then we averaged the high-resolution profiles over ~80 km distance along the track and over 1 km vertically. Besides testing noise-free simulations, we also checked the effects introduced by instrumental noise, which we estimated from the uppermost parts of measured profiles. For both instruments, these measurements are cloud-free and the molecular return is supposed to be smooth. Correspondingly, we estimated it by a least-square fit to measured molecular return and subtracted from the profile. The root-mean-square of the remaining difference gave us a noise level, which we used in the simulations. For CALIOP, the noise level obtained for instantaneous measurements was scaled in accordance with the averaging distance (see Section 3.3). Overall, we considered about 1E5 pairs of pseudo-collocated averaged profiles of $SR(532nm, z)$ and $SR'(532nm, z)$. Using these pairs and applying the same cloud detection threshold (Eq. 9), we

estimated the cloud fraction profiles (Eq. 11) and the occurrence frequency profiles for the simultaneous cloud detection by both instruments (Eq. 12). Finally, we estimated the normalized cloud detection agreement (Eq. 13).

**4.2. Horizontal and vertical averaging and its effects on ALADIN's capability to retrieve clouds**

A common way of reducing noise in the observations is accumulation of signal and averaging N realizations of the same measurement will reduce noise level by a factor of $\sqrt{N}$. The reverse side of this improved signal-to-noise ratio (SNR) is a loss of information if the signal varies. The Nyquist–Shannon–Kotelnikov sampling theorem says that "if a function x(t)
contains no frequencies higher than B hertz, it is completely determined by giving its ordinates at a series of points spaced 1/(2B) seconds apart". For satellite observations of clouds, we can reformulate this as follows: for a scene composed of a mixture of clear sky and clouds (e.g. Fig. 4a), the averaging length should be comparable to the size of the smallest element one wants to resolve. One can show that neglecting this rule will lead to overestimating cloud fraction. Consider an inhomogeneous scene, which is measured 250 times (line 1 in Fig. 4a), 80 of which give a strong backscatter signal from
clouds and 170 are clear sky cases. Here, the cloud fraction calculated in accordance with Eq. 11 will be equal to 0.32. At the same time, averaging the same scene through one long measurement (line 2 in Fig. 4a) will return CF=1.0 because $N_{cld} = 1$ will be triggered by a strong signal coming from a mixture of thick clouds and clear sky cases. This problem has been addressed before (e.g. Chepfer et al., 2010, 2013), and here we will illustrate it in application to CALIOP/ALADIN comparisons at their native scales to see what are the effects of averaging in ALADIN. In this exercise, we use two
resolutions of ALADIN. Even though the PRF of ALADIN laser is 50 Hz, its ACCD accumulates the backscattered photons over the periods of 0.4 s (~2.9 km) and transmits them to the ground where the L2A processor further averages the signal, leading to 87 km resolution of the L2A optical products. Recently, the onboard accumulation time (or distance) was doubled because of optical losses in the emission path of the instrument. In the exercise below, we show the estimates for 2.9 km averaging (line 3 in Fig. 4a), which corresponds to the period analyzed in this work.
For this exercise, we used the setup described in Sect. 4.1, picked up a piece of orbit, which contains both thick and thin, single- and multilayer clouds. The first simulation (Fig. 4b) shows the scattering ratio for this scene calculated at the original vertical and horizontal resolutions of CALIOP (60m and 333 m, respectively). In Fig. 4c, red color marks the areas, for which the scattering ratio is greater than cloud detection threshold (Eq. 9). Dashed circles mark the areas that require specific attention (see below). In Fig. 4d,e, we show the results of the same calculations performed at the resolution of ALADIN L2A
product (1 km V, 87 km H). As one can see, the clouds are reproduced well, except for the areas marked by dashed circles. In the first and second circles from the left, the cloud is not detected because it is thin and the signal is not strong enough to trigger the detection. However, in the third circle, the cloud fraction is overestimated. In Fig. 4f,g, we estimate the ALADIN cloud detection at the same vertical resolution of 1 km and horizontal resolution of 2.9 km. As one can see, Fig. 4g resembles Fig. 4c in encircled areas much more than Fig. 4e does. The thin cloud in the second dashed circle could be further improved
if the vertical resolution was different. Technically, one could improve it for certain heights at the cost of the other ones, but

practically the ALADIN vertical resolution is adjusted mostly for surface and boundary areas, and the total number of vertical bins is limited by 24, so we did not attempt to model this setup. Summing up, if the L2A processor of ALADIN manages to process real measurements at ~3km horizontal resolution, we would recommend this resolution as a tradeoff between information content and noise for this instrument. For future lidar missions, it is highly advisable to register data at CALIOP vertical and horizontal resolutions and average them only if needed to detect long thin cloud and/or aerosol layers.

### 4.3 Theoretically achievable cloud detection agreement between ALADIN and CALIOP

In Fig. 5, we show the profiles of $CF^{theor}(z)$, $CDA^{theor}(z)$, and $CDA_{norm}^{theor}(z)$ estimated in the approach outlined above. To address the contribution of different processes to the cloud detection agreement, we show both the simulations performed with the instrumental noise and diurnal variation and the simulations performed without these perturbations. As one can see, both the $CDA^{theor}(z)$ and $CDA_{norm}^{theor}(z)$ are mostly defined by a horizontal variability of aerosols/clouds combined with differences in viewing geometries of two instruments. Observation noise and diurnal variation play the secondary role (compare the curves with and without variations or "noise" in Fig. 5). Overall, we estimate the mean value of the theoretically achievable normalized cloud detection agreement $CDA_{norm}^{theor}(z)$ for the collocated data in the outlined setup to be equal to 0.81±0.07. As one can see, the vertical profile of $CDA_{norm}^{theor}(z)$ does not change much with altitude, indicating that the primary sources of discrepancy are the observation geometry and the spatial variability of clouds combined with the chosen collocation criterion. If the noise were the primary source of discrepancy, we would observe a decrease of $CDA_{norm}^{theor}(z)$ profile with height. In Sections 5.3 and 5.5, we will use the theoretical limit of 0.81±0.07 obtained in this section as a benchmark.

## 5. Analysis of the ALADIN and CALIPSO observations

### 5.1 Comparison of SR-height histograms in each latitude band

To give a general overview of the agreement between the $SR(532nm, z)$ and $SR'(532nm, z)$, we have split the collocated data to latitudinal zones: 90S−60S, 60S−30S, 30S−30N, 30N−60N, 60N−90N (Fig. 6). If the detection efficiency of different cloud types were the same for two instruments, the pairs of Fig. 6 panels (a;f), (b;g), (c;h), (d;i), and (e;j) would have been close to each other because of two reasons. First, the horizontal variability of clouds would have canceled out due to averaging over many profiles within the zone. Second, the diurnal variation is minor over oceans, which make up two-thirds of the data used for Fig. 6 (Noël et al., 2018; Chepfer et al., 2019; Feofilov and Stubenrauch, 2019). Analyzing the Fig. 6 one can note: (1) the SR-height histograms of CALIOP (Fig. 6c-e) show two distinct peaks corresponding to low-level and high-level clouds; this feature is coherent with other observations, e.g. with GEWEX (Global Energy and Water cycle Experiment) cloud assessment (Stubenrauch et al., 2013); (2) the SR-height histograms built for $SR'(532nm, z)$ retrieved

from ALADIN's observations (Fig. 6f-j) are characterized by a smoother occurrence frequency plot where the two-peak structure is less pronounced than in CALIOP; (3) even though ALADIN detects polar stratospheric clouds (PSCs), its overall sensitivity to clouds above ~3 km altitude is lower than that of CALIOP; (4) in each latitude band, the SR-height histograms agree reasonably well up to ~3km altitude; (5) both datasets show a layer of enhanced backscatter closer to the tropopause, which is not strong enough to trigger the cloud detection defined in this work. (6) the ALADIN's PSCs retrieved from $SR'(532nm, z)$ appear "brighter" than those estimated from CALIOP's $SR(532nm, z)$. This is likely an artefact due to our conversion of the ALADIN signal into $SR'(532nm, z)$ which assumes the particulate backscatter is about the same at 532 and 355nm (Eq. 8) while about 50% of PSCs contain droplets composed of super-cooled ternary solutions (STS) the backscatter of which at 355 nm is roughly twice as large as that of 532 nm (Jumelet et al., 2009). The latter reason explains ALADIN's higher sensitivity to PSCs compared to cirrus clouds. In the next step, we compare the "instantaneous" profiles provided by CALIOP and ALADIN having in mind the cloud detection sensitivity issues observed in Fig. 6.

## 5.2 Comparing individual SR profiles

Since Fig. 6 revealed certain differences between the two datasets, we inspected collocated data looking for the specific cases, which would explain the differences shown in Fig. 6. First, we wanted to test the ALADIN's capabilities of high cloud detection. The subset we used for this task had to satisfy the following criteria: (1) both instruments should have at least one strong SR peak; (2) the height of this peak detected by one instrument should match the height of the peak detected by a second instrument within 1 km; (3) the CALIOP SR profile should have a peak at or above 9 km (Fig. 7a-j). For the comparison purposes, the panels in Fig. 7 represent the individual profiles belonging to the same 5 zones as the panels of Fig. 6. As for the potential capability of ALADIN to detect high clouds, the subset shown in Fig. 6a-e represents the cases for which this instrument retrieved the peak of about the same magnitude and height as the peak detected by CALIOP. Even though these cases exist, they are less frequent than those shown in Fig. 7f-j when ALADIN misses a high cloud, but detects a lower cloud reported by CALIOP.

To test whether the said mismatch is linked with the diurnal variation, we varied Δtime in 3−12h limits, but this did not change the frequency of occurrence of high and low cloud detection. This gives a hint that the instrumental part itself provides the backscatter information sufficient for cloud detection up to 20 km, but the detection algorithm suppresses noisy solutions. The reasons for this presumable "noise" might be linked with instrumental issues discussed below, but they might be also related to the ratio of particulate and molecular backscatter at 355 nm. Let's have a closer look: the molecular signal is stronger at 355nm and the particulate signal is comparable to that at 532 nm. At the same time, ALADIN is an HSRL instrument, and the separation to molecular and particulate component requires disentangling of the signals measured in Mie and Rayleigh channels (cross-talk correction). Correspondingly, the error propagation in this procedure might adversely affect the SNR in Mie channel and, therefore, the SNRs of the extinction, backscatter, and recalculated $SR'(532nm, z)$. Characterizing these differences and their impact on retrieved clouds is beyond this study and it requires further investigation, but we believe that the high cloud detection agreement might be improved by studying the collocated cases

provided in the supplementary materials and by applying different noise filtering techniques in the L0→L1→L2 elements of the ALADIN retrieval chain. As for the Fig. 7k-o, we will discuss them below in the context of low-level cloud observations.

**5.3 Cloud detection agreement**

In Fig. 8, we show zonal cloud fraction profiles built from the collocated dataset of $SR(532nm, z)$ and $SR'(532nm, z)$ using the same threshold (Eq. 9) for both datasets. Despite the differences in SR absolute values, the $CF(z)$ profiles estimated from CALIOP and ALADIN demonstrate reasonable agreement. The Pearson's correlation coefficient for the panels in Fig. 8a,b varies between 0.7 and 0.9 for most heights (Fig. 8c) and the relative difference between the panels changes from 50% in the
lower layers through minus 50% at 11km and to 25% near tropopause (Fig. 8c).

To illustrate the zonal $CF(lat, z)$ profiles behavior, we split the collocated data into four groups defined in Section 3.5 (Eq. 12 and text preceding it). In Fig. 9, we show the distributions of $R_{YES\_YES}(lat, z)$, $R_{NO\_NO}(lat, z)$, $R_{YES\_NO}(lat, z)$, and $R_{NO\_YES}(lat, z)$. From the definition (Eq. 12), it follows that for an ideal agreement, the $R_{YES\_NO}(lat, z)$ and $R_{NO\_YES}(lat, z)$ values should be equal to zero. However, Fig. 9c and Fig. 9d show occurrence frequencies comparable to those of Fig. 9a.
From the study presented in Section 4.2, we expect that the ratio of $R_{YES\_YES}(lat, z)$ to $CF(lat, z)$ should be about 0.81±0.07 if we take CALIOP as a reference for cloud detection sensitivity. If we build the $CDA_{norm}(z)$ estimated from Fig. 9a and Fig. 8 (Eq. 13), we see that it fits the prescribed value up to ~3km (cyan curve in Fig. 5). Above this altitude, the normalized cloud detection agreement oscillates around 0.5.

The distribution of $R_{YES\_YES}(lat, z)$ (or $CDA(lat, z)$) shown in Fig. 9a resembles a typical cloud fraction profile plot
(compare with Fig. 8). This is not surprising because $R_{YES\_YES}(z)$ must turn to $CF(z)$ if the agreement is perfect (see Eqs. 11, 13). Even though the distribution in Fig. 9a looks physical, the ratios for the heights above 3km are ~40% lower than expected from the theoretical estimates (see Fig. 5 or compare Fig. 9a with Fig. 9c). As one can see from Fig. 9c, the missing cases also form a structure, which resembles $CF(lat, z)$ distribution. This shows that 40% of ALADIN's $SR'(532nm, z)$ values are below the threshold (Eq. 9). Technically, we could fix this by lowering the detection threshold, but this would
increase the $R_{NO\_YES}(lat, z)$ occurrence frequency (Fig. 9d) that is not desired.

As for $R_{NO\_NO}(lat, z)$ shown in Fig. 9b, it is close to 100% in the high-altitude area where there are no clouds. This indicates that the false cloud detection induced by a small SNR in cloud-free area is rare for both instruments. We consider this to be a good sign as it shows the stability of the ALADIN retrieval algorithm for weak signals.

We draw the readers' attention to the fact that we did not expect the NO_YES mismatches for the considered combination of
lidars at low altitude (Fig. 9d). Let us explain. The molecular extinction at 355 nm is larger than at 532 nm and the observation geometry of ALADIN makes the optical paths *1 / cos(35°) = 1.22* times longer than those for CALIOP, where 35° is a satellite viewing angle. The particulate backscatter coefficients at these wavelengths are almost the same. Therefore, for the same low-level cloud, all other factors being equal, cloud detection should be more probable for CALIOP and not for ALADIN. The typical individual profiles corresponding to NO_YES mismatches are shown in Fig. 6k-o. As one can see,

despite the unfavorable observation conditions (e.q. a cloud with peak $SR(532nm)$ value of ~20 at 7 km in Fig. 7 l), ALADIN retrieves one or two valid points beneath a cloud detected by both instruments. Let us consider plausible reasons for the observed behavior:

(1) Since many cases of NO_YES type are over the ocean, one can rule out the continent surface echo contamination of the backscattered signal at 2 km height.

(2) The horizontal cloud inhomogeneity could explain the individual cases shown in Fig. 7k-o, but it cannot explain the general behavior observed in Fig. 9d.

(3) The higher detection rate in the lower layers cannot be fixed by increasing the SR threshold (Eq. 9) because it will adversely affect the agreement at other altitudes.

(4) Since the $SR'(532nm, z)$ values in this work were recalculated from the source ALADIN data at 355 nm, the
uncertainties and biases of the parameters used for recalculation (Section 3.4) could have biased the results. These effects would accumulate along the line of sight, so one can expect the errors to be larger near the ground.

Let us verify the last hypothesis and consider the elements of Eq. 1:

(4a) $\alpha_{part}(\lambda, z)$ and $\beta_{part}(\lambda, z)$ are retrieved from ALADIN and their uncertainty or bias will propagate through the calculations. Moreover, a small bias in $\alpha_{part}(\lambda, z)$ will accumulate with distance (Eq. 1). Therefore, one cannot
rule out this source of discrepancy. To explain the observed behavior, $\alpha_{part}(\lambda, z)$ should be biased towards smaller values, or $\beta_{part}(\lambda, z)$ should be biased towards larger values in the considered layers.

(4b) $\alpha_{mol}(\lambda, z)$ and $\beta_{mol}(\lambda, z)$ are calculated with high accuracy given that the molecular density profile in CALIOP comes from the GEOS-5 DAS database, see (Rienecker, 2008). The uncertainties of the parameters used for their estimate are small (Bucholz, 1995; Ciddor, 1996). Therefore, it is unlikely that they can explain the observed
NO_YES cases.

(4c) The physical meaning of the multiple scattering coefficient η is an increase in number of photons remaining in the lidar receiver field of view (Garnier et al., 2015). Its value depends on type of scattering media and FOV of the lidar and varies between 0.5 and 0.8 for commonly used lidars (Chiriaco et al., 2006; Chepfer et al., 2008, 2013; Garnier et al., 2015). For ALADIN, with its narrow FOV of ~0.02 mrad, it is usually considered to be equal to 1,
but, as we explained, we used η=0.9 for the conversion. Decreasing of the η value is not justified; in addition, it will increase the number of low-level clouds and NO_YES cases. On the other hand, its increase will reduce the fraction of NO_YES cases in the lower layers, but it will worsen the YES_YES agreement at the same time. Still, this parameter remains on the list of the variables, which may affect the quality of the $SR'(532nm, z)$ conversion, and adapting the most recent model of η for ALADIN should be the next step in merging the cloud record from
these two lidars.

Summarizing this section, we conclude that (a) a cloud layer detected by CALIOP is detected by ALADIN in ~80% of cases for cloud layers below ~3km and in ~50% of cases for higher cloud layers; (b) in the cloud-free area, the agreement between

the datasets is good, indicating the stability of ALADIN L2 retrieval algorithm for weak signals; (c) half of the cases when ALADIN detects a low-level cloud missed by CALIOP cannot be explained by sampling and geometrical differences, diurnal variation, or uncertainties in the $SR'(532nm, z)$ profile recalculation.

## 5.4 Cloud altitude detection sensitivity

We now analyze if clouds detected by the two lidars peak at the same altitude. We note that we are not looking for an altitude offset here. The altitude detection of both instruments is beyond question. Instead, we would like to check whether the higher detection rate of lower clouds leads to slight systematic differences in the cloud altitudes derived from the 2 lidars. To do so, we have carried out the following analysis. For each pair of collocated profiles selected for YES_YES plot (Fig. 9a), we scanned vertically through ALADIN profile step by step, looking for a local maximum, satisfying the following conditions:

$$SR(i) > 5; \; SR(i) > SR(i-1); \; SR(i) > SR(i+1) \tag{13}$$

For each local peak found, we have searched for a peak or for a maximal value of CALIOP's SR(z) profile in the vertical vicinity of ±3 km from the peak height determined from ALADIN. We have chosen these search limits by inspecting the collocated profiles, considering the natural variability of cloud heights at distances similar to those used in collocations. According to our analysis of CALIOP data, at these distances ~75% of clouds move vertically by less than 1 km, ~8% by 1−2km, ~5% by 2−3km, ~4% by 3−4km, ~3% by 4−5 km and ~5% by more than 5 km. We note that by imposing the ±3 km search criteria we filter out about 12% of the cases linked to natural variability that slightly reduces the number of cases selected for the analysis. At the same time, we lower the rate of picking up the peak from a different cloud layer.

We stored the differences between the ALADIN's and CALIOP's cloud peak heights and then averaged them in the corresponding latitude/altitude bins (Fig. 10). As one can see, the agreement is good for the tropical high clouds. This is probably linked with thick Ci clouds, which should be reliably detected by both instruments. For the Southern polar zone, this figure reveals the PSCs, which are barely visible in Fig. 9a, but which can be seen in Fig. 6f for ALADIN. These clouds form at very low temperatures and are partially composed of large ice particles yielding a reflection detected at both wavelengths if the layer is thick enough (e.g. Adriani et al., 2004; Noël et al., 2008; Snels et al., 2021). However, the $SR'(532nm, z)$ in PSC are likely positively biased due to our conversion of ALADIN data to $SR'(532nm, z)$ as discussed in Sect. 5.1.

As one can see in Fig. 10, the higher sensitivity to low-level clouds shifts the average ALADIN's cloud height downwards compared to CALIOP. At the heights of 3−5 km, the shift is as large as 0.8−1.2 km. One can attribute a part of this effect to the reasons discussed for the existence of NO_YES cases (e.g. if one assumes larger values of η, the average downward shift will be smaller, but this kind of "tweaking" would need to be justified). Summarizing, the assumption of skewing the average cloud height through higher sensitivity to lower clouds proves to be valid, and we estimate a mean downward shift to be equal to 0.5±0.6 km.

## 5.5 Temporal evolution of cloud detection agreement

As mentioned in Section 2.1, the ALADIN lidar faced several technical issues, which hindered getting the planned specifications. Among them, we named the "hot pixels" issue, which requires some explanation. First, the information from them is not completely lost, and there is a way of recalibrating of these pixels (Weiler et al., 2021). Second, if we compare the hot pixels distribution for Mie and Rayleigh channel ACCD detectors for the period considered in this work (see Table 2 of Weiler et al., 2021), we will see 3 and 5 new hot pixels for Mie and Rayleigh matrices, respectively. For Mie detector matrix, the lowermost hot pixel, which appeared during the considered period, corresponds to ~15 km height. Even though these pixels do not overlap with the maxima of cloud height distributions, they still might affect the retrieval results below because of the optical path passing through the corresponding layers (see Eq. 1). As for new Rayleigh hot pixels, the lowermost two correspond to 1 km height, the next two – to 5 km, and the last one – to 18 km. The Rayleigh matrix pixels are not directly linked to cloud detection, but their cross-talks are used in ALADIN's $\alpha_{part}(\lambda, z)$ and $\beta_{part}(\lambda, z)$ calculations, so they might also affect the results.

In Fig. 11a-d, we show the temporal evolution of $R_{YES\_YES}(time, z)$, $R_{NO\_NO}(time, z)$, $R_{YES\_NO}(time, z)$, and $R_{NO\_YES}(time, z)$ over the whole period of collocated data set (28/06/2019−31/12/2019). Figure 11e and Fig. 11f show the temporal evolution of $CDA_{norm}(time, z)$ in two forms: as a color plot and as 2D linear fitting at the heights characterized by high occurrence frequency (0.75 km, 2.25 km, 6.25 km, 8.75 km, and 10.0 km). Unfortunately, the period available for analysis does not cover the entire year, so the plots Fig. 10a-d can be affected by seasonal variation of cloud distributions. Still, the latitudinal and longitudinal coverage of collocated data does not change throughout the year and a mixture of Northern and Southern hemispheres should partially compensate for seasonal anomalies. As for the $CDA_{norm}^{obs}(time, z)$ panels (Fig. 11e,f), the normalizing by $CF(time, z)$ should compensate for the seasonal variation in these plots. Possible artefacts linked to laser power degradation, hot pixels, and bias correction would likely show up as a decrease in $R_{YES\_YES}(time, z)$ and $R_{NO\_NO}(time, z)$ occurrence frequencies (Fig. 11a,b) and as an increase in $R_{YES\_NO}(time, z)$ and $R_{NO\_YES}(time, z)$ occurrence frequencies (Fig. 11c,d).

However, this is not the case: visually, all panels of Fig. 11a-d do not show any anomaly, which would go beyond their noise levels. We note that there is a special region corresponding to a forced vertical bin size reduction in the period of 28/10/2019−10/11/2019, which is marked by white dashed lines in Fig. 11 and which should not be considered at heights below 2250m. To quantify the tendencies and to compare them with noise levels, we analyzed the $CDA_{norm}^{obs}(time, z)$ distributions (Fig. 11e,f). The results presented in these panels confirm the previous conclusions regarding the $CDA_{norm}^{obs}(z)$ profile: for the clouds below 3 km, it is better than for higher ones (68±12% at 0.75 km and 55±14% at 2.25 km versus 34±14%, 39±13%, and 42±14% at 6.75 km, 8.75 km, and 10.25 km, respectively. The uncertainty limits in these estimates are relatively large. Nevertheless, the absence of statistically significant trends indicates that the compensation for hot pixels effects (Weiler et al., 2021) and for signal losses in the emission and reception paths removes the signatures of the experimental issues from the ALADIN L2 optical products at least down to these uncertainty limits.

## 6. Conclusions

The active sounders are advantageous for atmospheric and climate studies because they provide precise vertically resolved information. Building a long-term cloud profile climate record with the help of these instruments requires understanding the differences between space-borne lidars operating at different wavelengths, flying on different orbits and using different observation geometries, receiving paths, and detectors. In this article, we compared the ALADIN and CALIOP lidars using their scattering ratio products (CALIPSO-GOCCP and Aeolus L2A, Prototype_v3.10) for the period from 28/06/2019 to 31/12/2019. We defined the spatial collocation criterion of $1°$ based on the averaging distance of Aeolus L2A Prototype_v3.10 data. The temporal collocation criterion of $\Delta time < 6h$ used in this work is a tradeoff between the geographical coverage of the collocated profiles, their number, and uniformity of $\Delta time$ distribution throughout the globe. With the named criteria, we found ~7.8E4 collocated nighttime profiles, which underwent a series of analysis summarized here.

For an adequate comparison with the CALIOP's $SR(532nm, z)$, we converted ALADIN's $\alpha_{part}(355nm, z)$, $\beta_{part}(355nm, z)$, and $SR^A(355nm, z)$ to $SR'(532nm, z)$ and we discussed the uncertainties of this conversion.

Before analyzing the actual observations, we performed a numerical experiment to estimate the best achievable cloud detection agreement between the two missions. We found that the agreement between ALADIN and CALIOP clouds should be about $0.81\pm0.07$, regardless of the altitude. The numerical experiment used the outputs from a global atmospheric model coupled with a lidar simulator, a horizontal cloud variability parameterization, and considering the lidar orbit, sampling, averaging, noise, and observation geometry differences in the two lidars.

Analyzing the actual observations, namely, the ALADIN dataset converted to $SR'(532nm, z)$ profiles and compared with the $SR(532nm, z)$ profiles of CALIOP both at a vertical resolution of 1km and horizontal resolution of 80km, we report a good agreement in the lower atmospheric layers. Above 3 km, the agreement is worse. We explain this by lower SNR for ALADIN at these heights that is due both to physical reasons (ratio of particulate to molecular backscatter is smaller at 355 nm than at 532 nm) and technical reasons (lower emission and lower transmissivity of receive path than planned). The PSC detection by ALADIN confirms this hypothesis: at 355 nm, the PSC backscatter is stronger than at 532 nm that leads to high SNR and reliable cloud retrieval.

Switching from the absolute $SR(532nm, z)$ and $SR'(532nm, z)$ values to cloud fraction profiles obtained by applying a fixed cloud detection threshold of $SR > 5$, the zonal mean cloud profiles of the two compared instruments show relatively good agreement, with Pearson's correlation coefficient varying from 0.7 to 0.9 and relative difference varying within $\pm50\%$ on the altitude. In the lower 3 km, the estimated $CDA_{norm}^{obs}(z)$ profile almost reaches its theoretically estimated value $CDA_{norm}^{theor}$ of $0.81\pm0.07$ whereas in the upper layers, its value is about 40% less. Better detection of lower clouds skews the mean ALADIN's cloud peak height in pairs of ALADIN/CALIOP profiles by ~$0.5\pm0.6$ km downwards. For the reasons explained above, the agreement of PSC peak heights and of tropical high clouds does not suffer from these effects. In the

cloud-free area, the agreement between two instruments is good, indicating a low rate of noise-induced false detection for both instruments.

Last, but not least, the temporal evolution of cloud agreement does not reveal any statistically significant change during the considered period. This shows that hot pixels and laser energy and receiving path degradation effects in ALADIN have been mitigated at least down to the uncertainties of the following normalized cloud detection agreement values: $68\pm12\%$,

$55\pm14\%$, $34\pm14\%$, $39\pm13\%$, and $42\pm14\%$ estimated at 0.75 km, 2.25 km, 6.75 km, 8.75 km, and 10.25 km, respectively. We believe that the provided collocated dataset will facilitate the further analysis and improvement of ALADIN L2A data. From our point of view, the outlook for a cloud product retrieved from ALADIN observations to be part of cloud lidar long record is promising: its L1 to L2 algorithms and the thresholds can be adapted to retrieve some of the same clouds as from CALIOP. This will help to better understand the instrumental and observational differences and build a long-term cloud

profile climate record.

### Data availability

The collocated dataset used in this work can be downloaded from ResearchGate repository using the following link https://doi.org/10.13140/RG.2.2.16562.94409 (Feofilov et al., 2021)

### Author contribution

HC, VN, MC, and AF: conceptualization, investigation, methodology, and validation; RG, CG, PLM, and AF: data curation and formal analysis; AF: writing original draft; AF and HC: review and editing.

### Competing interests

The authors declare that they have no conflict of interest.

### Disclaimer

The presented work includes preliminary data (not fully calibrated/validated and not yet publicly released) of the Aeolus mission that is part of the European Space Agency (ESA) Earth Explorer Program. This includes aerosol and cloud products, which have not yet been publicly released. The processor development, improvement and product reprocessing preparation are performed by the Aeolus DISC (Data, Innovation and Science Cluster), which involves DLR, DoRIT, ECMWF, KNMI, CNRS, S&T, ABB and Serco, in close cooperation with the Aeolus PDGS (Payload Data Ground Segment).

**Acknowledgements**

This work is supported by the Centre National de la Recherche Scientifique (CNRS) and by the Centre National d'Etudes Spatiales (CNES) through the Expecting Earth-Care, Learning from A-Train (EECLAT) project. The processor development, improvement and product reprocessing preparation are performed by the Aeolus DISC (Data, Innovation and Science Cluster), which involves DLR, DoRIT, ECMWF, KNMI, CNRS, S&T, ABB and Serco, in close cooperation with 600 the Aeolus PDGS (Payload Data Ground Segment). P.-L. M. acknowledges supports from the U.S. Department of Energy, Office of Science, Office of Biological and Environmental Research, Regional and Global Model Analysis (RGMA) program area. The authors want to thank F. Ehlers (EOP-SMA/ESTEC/ESA), A. Straume (ESTEC/ESA), and O. Reiterbuch (DLR) for their comments on the preliminary version of the manuscript and three anonymous reviewers for their in-depth analysis and comments throughout the whole review process.

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

| Instrument | Orbit inclination [deg] | Equator crossing LT [h] | Off-nadir angle [deg] | PRF [Hz] | Native resolution [m] | L2 resolution resolution [m] |
|---|---|---|---|---|---|---|
| ALADIN | 96.97 | 6:00 / 18:00 | 35 | 50.0 | 140 (H) x 250-2000 (V) | 87000 (H) x 250-2000 (V) |
| CALIOP | 98.00 | 01:30 / 13:30 | 3 | 20.1 | 333 (H) x 60 (V) | 333 (H) x 500(V) |

**Table 1: Comparison of orbital parameters, viewing geometries, and resolutions of ALADIN and CALIOP instruments**

| $\Delta$time [h] | Daytime ×1E3 | Night-time ×1E3 | Total ×1E3 |
|---|---|---|---|
| < 1 | 4.1 | 3.4 | 7.5 |
| < 4 | 25 | 50 | 75 |
| < 6 | 90 | 77 | 167 |
| < 9 | 120 | 108 | 228 |
| < 12 | 133 | 115 | 248 |
| < 24 | 173 | 144 | 317 |

**Table 2: Number of collocated cases for Δdist < 1º and different Δtime values**

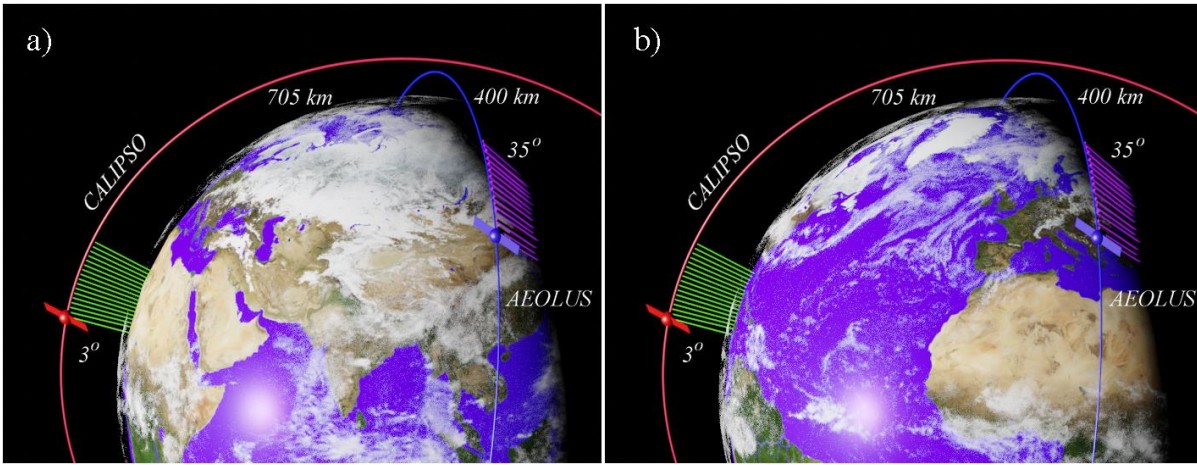

**Figure 1: Observation geometry and orbits of ALADIN/Aeolus and CALIOP/CALIPSO space borne lidars. ALADIN observes the atmosphere at dawn-dusk, whereas CALIOP passes the equator at 01:30 and 13:30 local solar time. The difference between (a) and (b) panels is in the position of Earth and the time: in (b), AEOLUS overflies the same area (centered over Africa) as was observed by CALIOP ~4.5 h earlier (in (a)).**

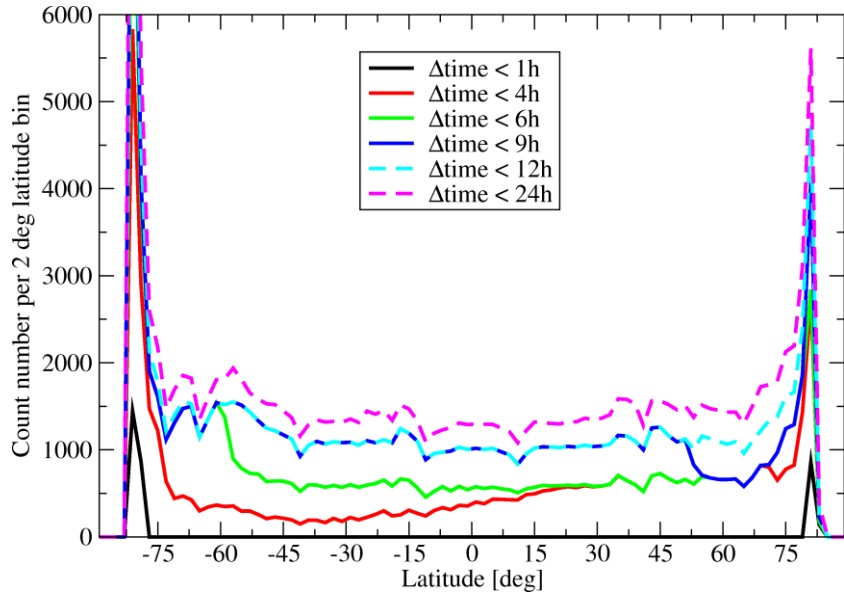

**Figure 2: Latitudinal coverage of collocated points for Δdist < 1° and different limits for Δtime.**

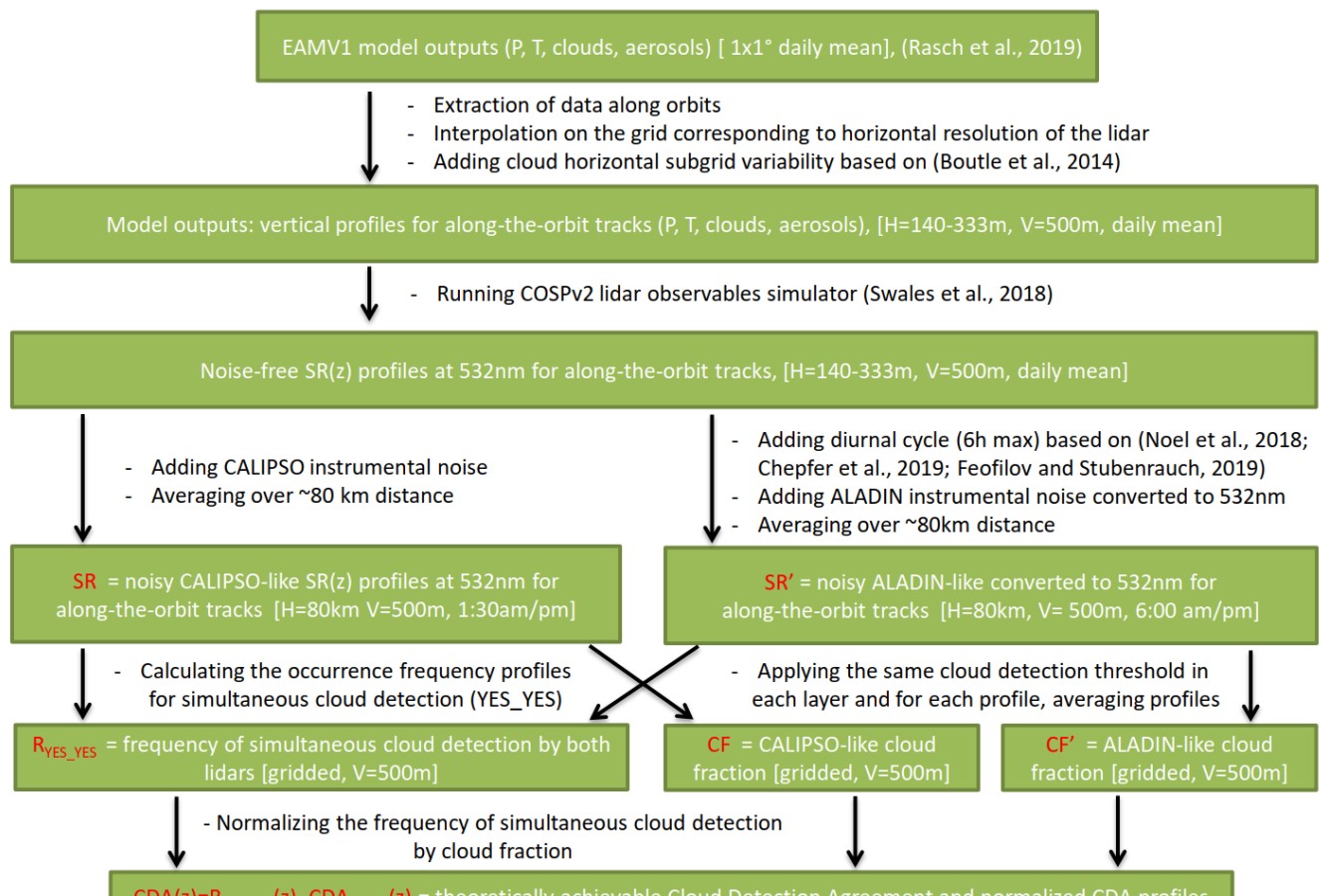

**Figure 3: A flowchart explaining the numerical experiment on estimating the best possible cloud detection agreement for a combination of ALADIN and CALIOP observations. Green boxes list the input and output data. Black text between boxes describes actions performed on each dataset. Red text in the boxes marks the datasets used in the estimation. White text in square brackets in the boxes indicates horizontal (H) and vertical (V) resolutions of the datasets.**

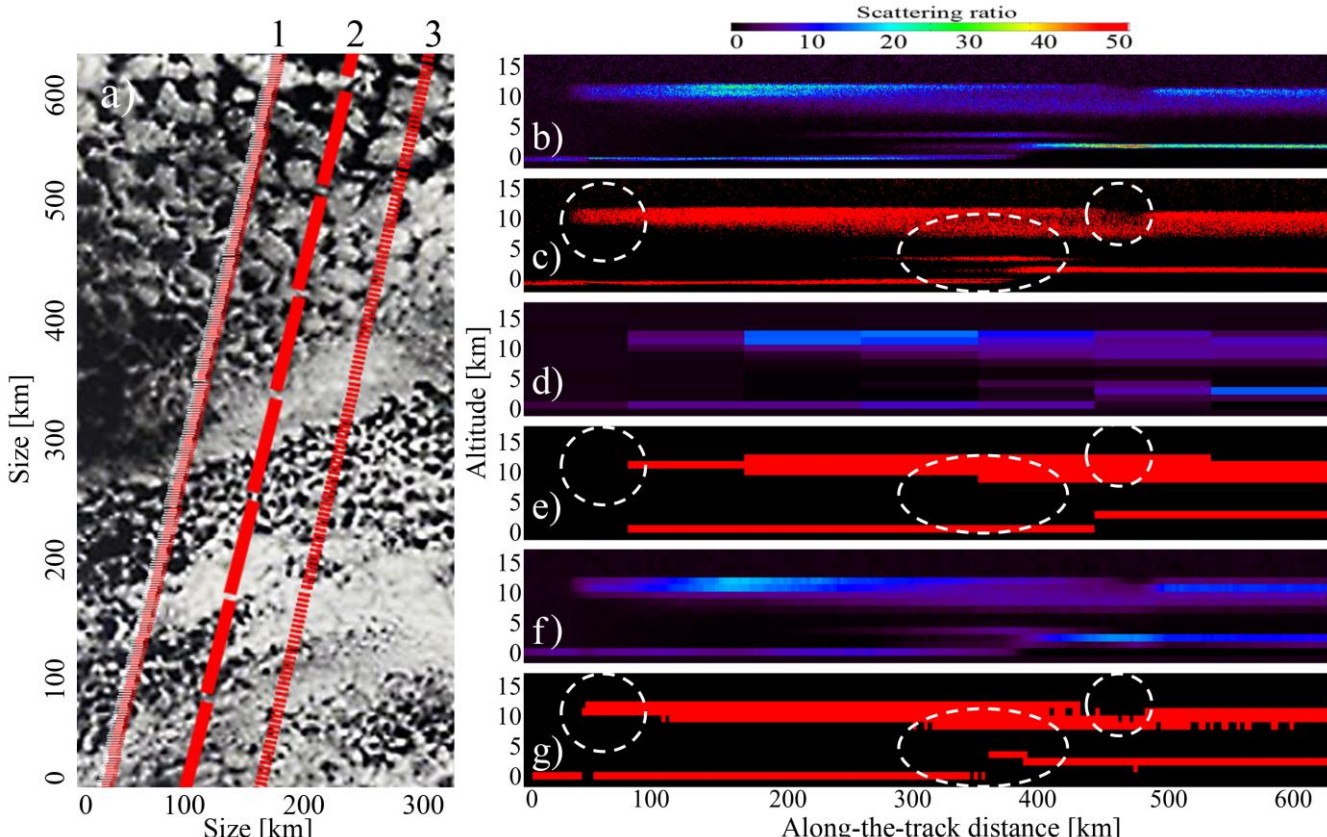

**Figure 4: Effects of horizontal and vertical averaging on cloud detection. (a) Visible satellite image from the NASA Moderate Resolution Imaging Spectroradiometer (MODIS) showing horizontal cloud structure of the order of 5-7km; image from the NASA MODIS Rapidfire archive; image scale is approximately 330 km x 660 km; the artificial CALIPSO and Aeolus tracks are superimposed to illustrate the problem of averaging over long distances; 1 – CALIOP at 333 m, 2 – ALADIN at 87 km, 3 – ALADIN at 2.9 km; (b) simulated scattering ratio for CALIOP at its native resolution of 333 m (H) and 60 m (V); (c) CALIOP clouds; (d) simulated scattering ratio for ALADIN converted to SR_532 at 1 km (V) and 87 km (H); (e) ALADIN clouds estimated for d; (f) same as d, but for 2.9 km (H); (e) ALADIN clouds estimated for f.**

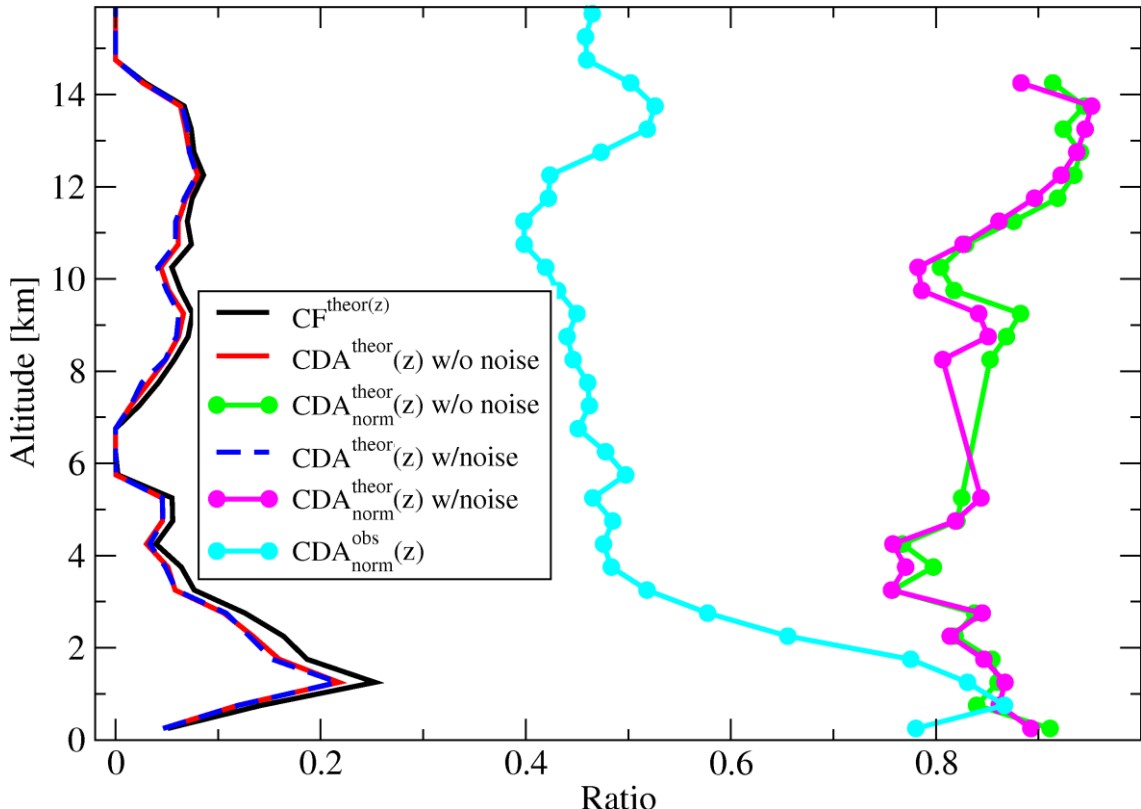

**Figure 5: Estimating theoretical cloud detection agreement (CDA) using pseudo-collocated $SR(532nm, z)$ and $SR'(532nm, z)$ profiles calculated using COSP2 lidar simulator coupled with the output of the EAMv1 atmospheric model (see Fig. 3). The definitions of cloud fraction (CF), CDA, and CDA$_{norm}$ variables are given in Section 3.5. "Noise" stands for calculations considering experimental noise of CALIOP and ALADIN and diurnal variation of the clouds for the collocation Δtime up to 6 h (see Fig. 2). The "$CDA_{norm}^{obs}(z)$" cyan curve comes from the analysis of real collocated data and is mentioned in Section 5.3.**

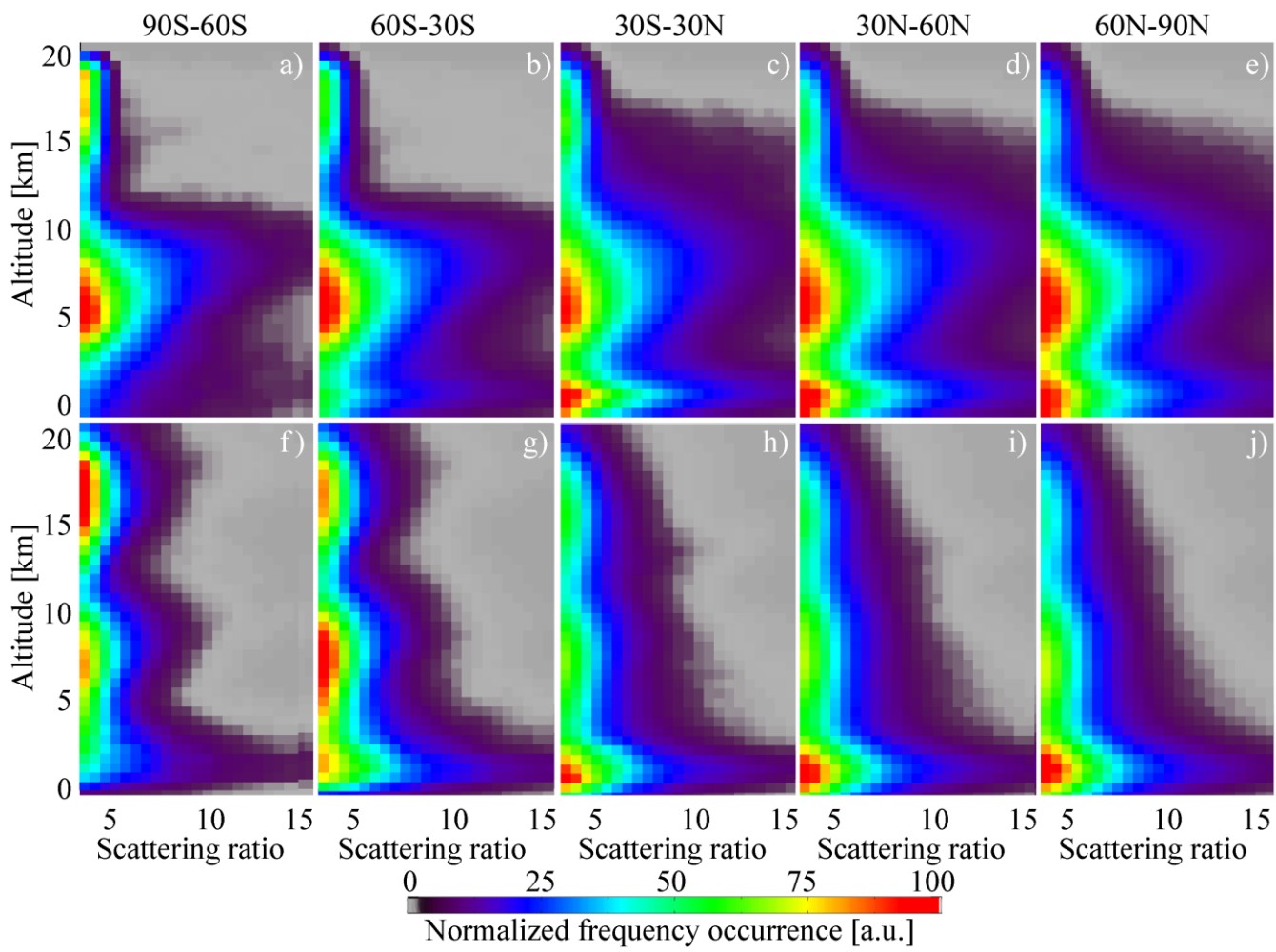

**Figure 6: Scattering ratio / height (SR-height) distributions in different latitude zones for the Δtime < 6h, Δdist < 1° collocated nighttime data subset (see Table 2): (a)-(e) CALIOP $SR(532nm, z)$ averages; (f)-(j) $SR'(532nm, z)$ estimated from ALADIN extinction and backscatter coefficients; (a,f) 90S-60S; (b,g) 60S-30S; (c,h) 30S-30N; (d,i) 30N-60N; (e,j) 60N-90N.**

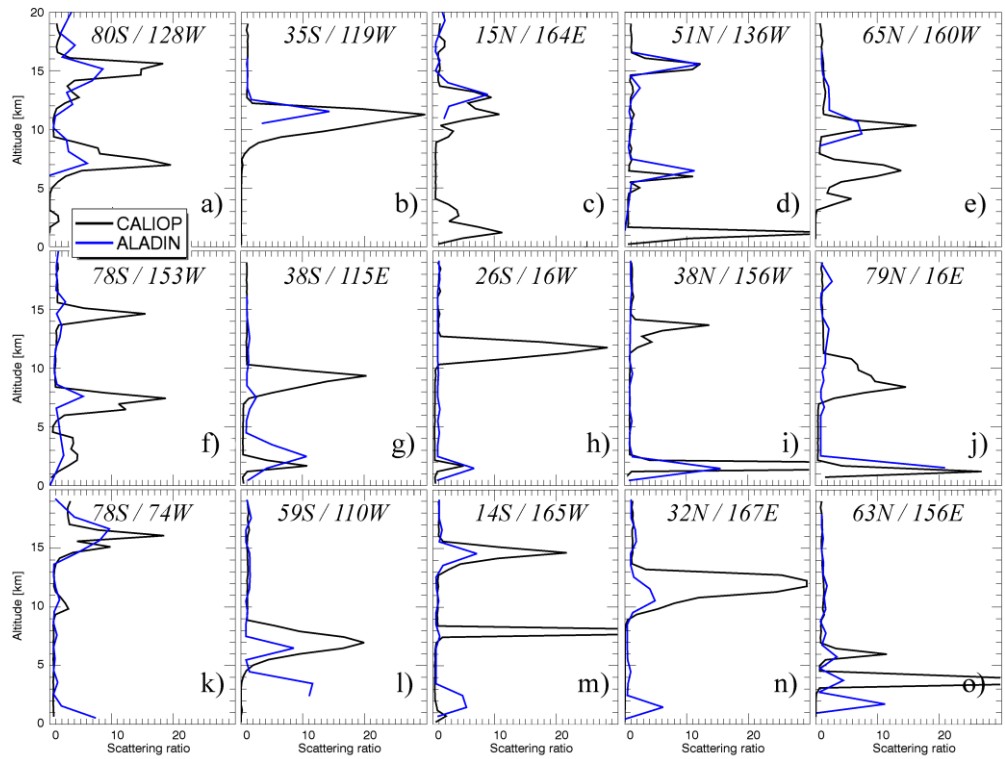

**Figure 7: Pseudo-instantaneous comparisons of collocated ALADIN L2A SR profiles and CALIOP SR profiles averaged over 67 km along the track: (a, f, k) 90S-60S; (b, g, l) 60S-30S; (c, h, m) 30S-30N; (d, i , n) 30N-60N; (e, j, o) 60N-90N; (a-e) cases confirming ALADIN's capability to detect high-level clouds; (f-j) cases showing the cases when ALADIN misses a high cloud detected by CALIOP; (k-o) cases showing a low level cloud detected by ALADIN and not detected bu CALIOP in the presence of a higher thick cloud detected by both instruments.**


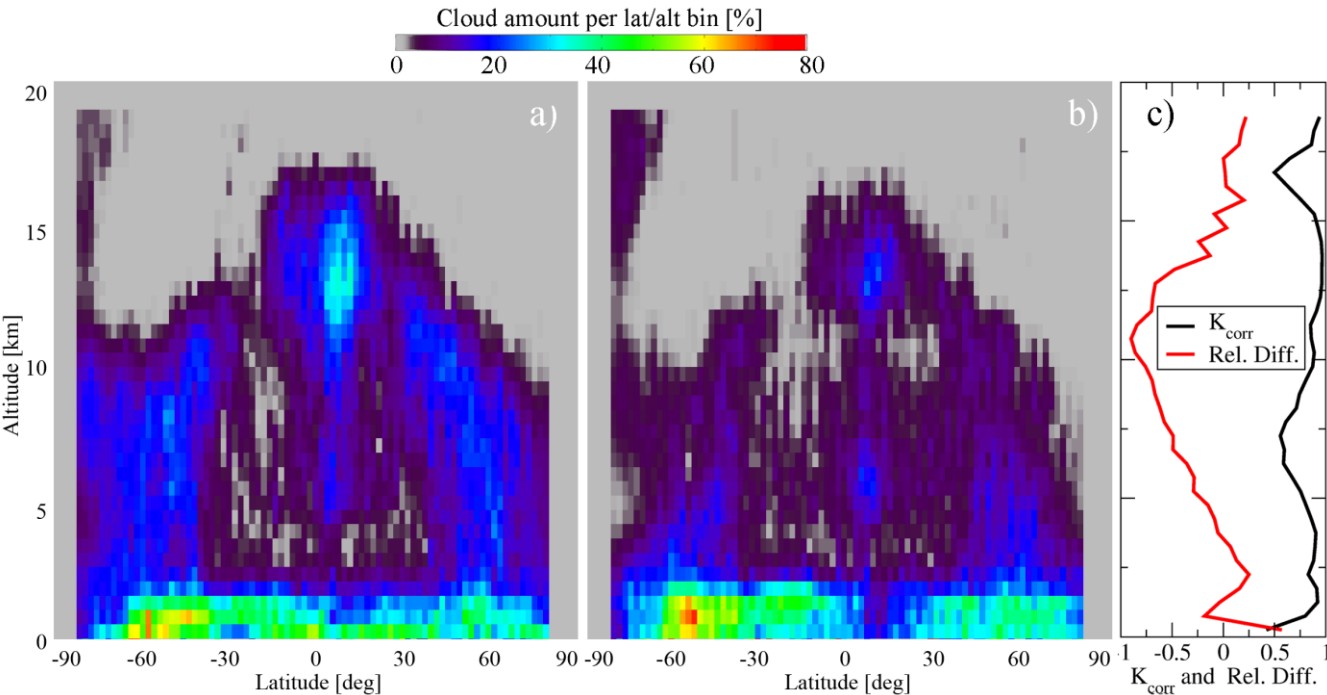

**Figure 8: Latitudinal/altitudinal distributions of cloud amount defined from (a) CALIOP and (b) ALADIN, and altitudinal profiles of Pearson's correlation coefficient and relative difference between ALADIN and CALIOP (c).**


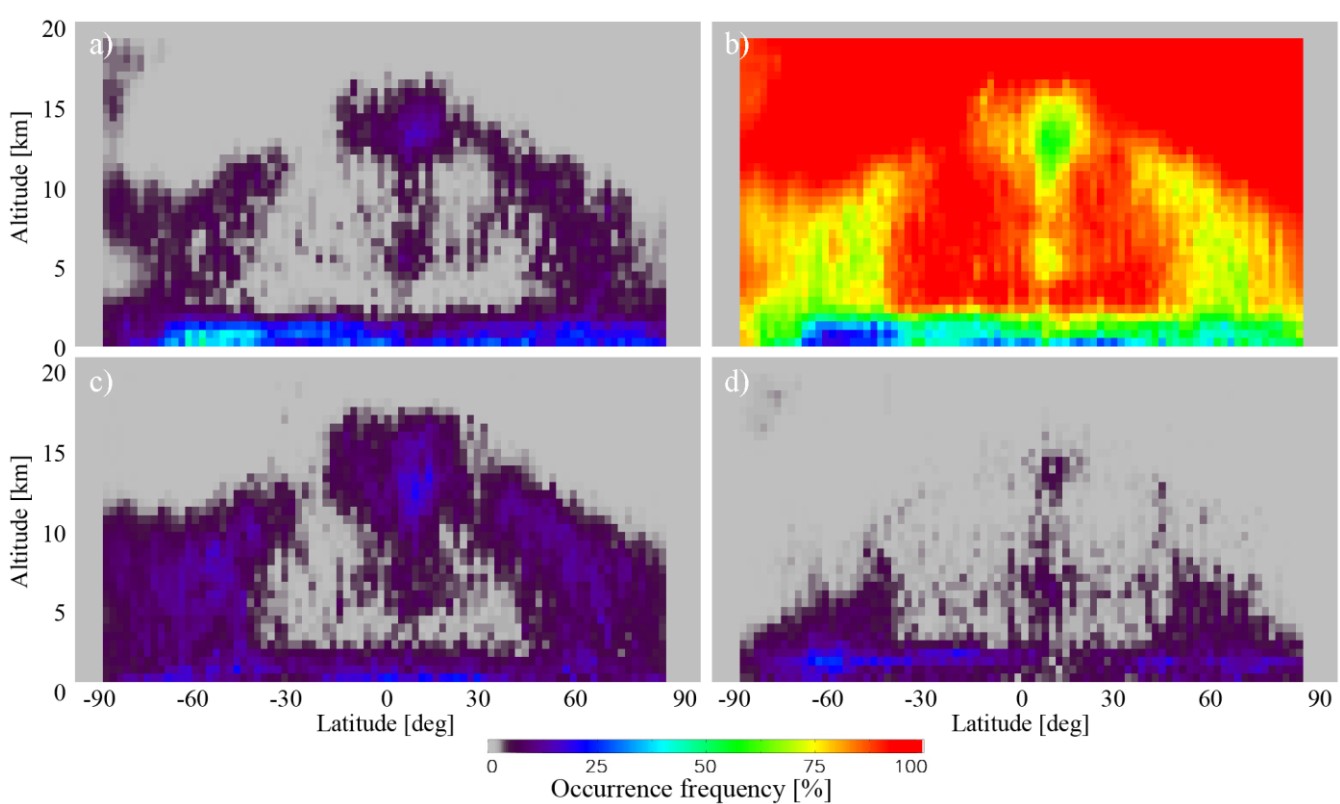

**Figure 9: Occurrence frequency for collocated observations: a) both CALIOP and ALADIN detected a cloud ($R_{YES\_YES}(lat, z)$); b) neither CALIOP nor ALADIN detected a cloud ($R_{NO\_NO}(lat, z)$); c) CALIOP detected a cloud, whereas ALADIN missed a cloud ($R_{YES\_NO}(lat, z)$; d) CALIOP missed a cloud, whereas ALADIN detected a cloud ($R_{NO\_YES}(lat, z)$).**

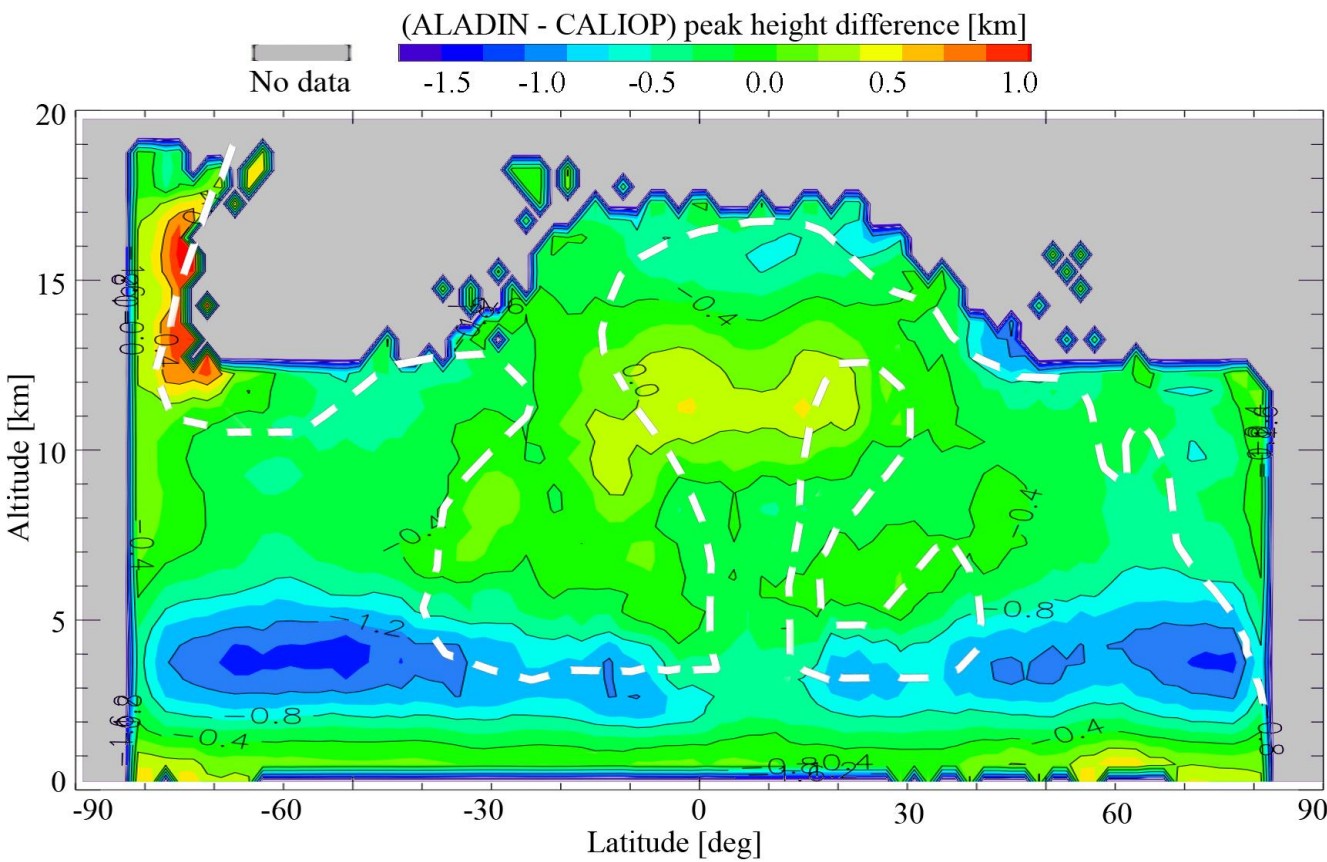


**Figure 10: Cloud altitude detection sensitivity represented as a height difference between the CALIOP local peak height and corresponding ALADIN's cloud peak height or maximal SR height found in the ±3 km vertical vicinity of CALIPSO peak. The subset corresponding to YES_YES selection (Fig. 9a) was used. White dashed isoline corresponds to colored area in Fig. 9a (occurrence frequency of about 5% and higher).**


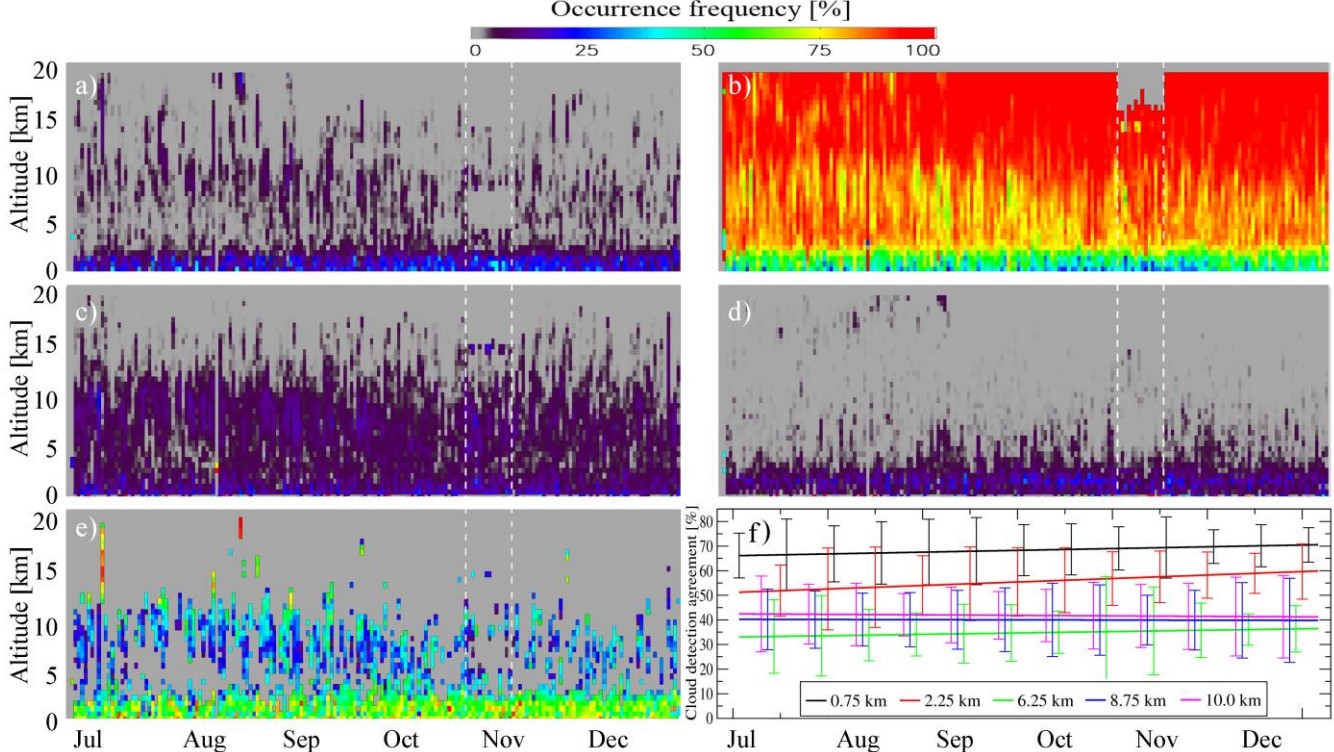

Figure 11: Temporal evolution of occurrence frequencies for a) $R_{YES\_YES}(z, time)$; b) $R_{NO\_NO}(z, time)$; c) $R_{YES\_NO}(z, time)$; d) $R_{NO\_YES}(z, time)$ for the period of 28/06/2019-31/12/2019. The legend is consistent with that of Fig. 9. White vertical dashed lines correspond to the Air Motion Vector (AMV) campaign (28/10/2019−10/11/2019), which is characterized by smaller bin sizes and, therefore, larger SNRs for Mie and Rayleigh channels up to the height of 2250m; e) normalized cloud detection agreement $CDA_{norm}(z, time)$; f) same as (e) presented for 5 heights as linear fits in 2D with error bars. The error bars were estimated as root-mean-square values for 1-week chunks of altitude subsets.