# Peer review of "Comparison of scattering ratio profiles retrieved from ALADIN/Aeolus and CALIOP/CALIPSO observations and preliminary estimates of cloud fraction profiles"

_Atmospheric Measurement Techniques, 2021_

## Referee Comment (RC1)

The paper deals with comparing the CALIOP and AEOLUS climatological backscatter data (half year in 2019). While the idea is good and has a high potential, the methodology and the presentation style are poor.

The wavelength difference between the two different lidars in space is only poorly accounted for, using an empirical 50-years old formula. Here, the authors should have used temperature and pressure profiles from NWP (as e.g. provided with Aeolus data) to calculate the molecular backscatter in the UV and visible range (i.e. at 532 nm) to make a real conversion of the scattering ratio and compare apples to apples.

Besides that, I also have the feeling that misinterpretation of Aeolus data is done while not taking into account the high contribution of molecular backscatter to the scattering ratio (see for example specific statement below under 7.).

 Thus, the manuscript suffers from a significant methodological weakness and any conclusion drawn from the current applied methodology is very questionable.

Furthermore, the presentation style needs to be improved. The language is hard to read and sometime really not understandable. E.g., already the abstract is hard to understand.

Also, the title does not at all reflect the content of the paper. Furthermore, I could not follow some of the argumentations. Often, statements are made without justification.

Therefore, I recommend the rejection of the manuscript, while at the same time encouraging the authors to re-submit a paper once the methodology and presentation style has been significantly improved.

Some comment general comments below, more detailed comments are given in the commented manuscript.

**Major comments:**

1. Title:

The title does not reflect the content of the paper. In fact, the authors focus only on the cloud detection capability based on scattering ratios. Furthermore, the whole instruction deals only with clouds and not a single word about scattering ratios is written.

2. Definition of scattering ratio

The scattering ratio which is the essential part of this manuscript has never been properly defined. According to the reference which is given, I assume that, "the ratio between the total backscatter by particles and molecules and the molecular backscatter" (according to Flamantm 2008) is meant, i.e. the ratio between the total backscatter (represented by particles and molecules) **to** the molecular backscatter.

**3. Wavelength conversion**

The conversion the authors use to account for the different wavelengths of CALIOP and AEOLUS is poor.
For example, I have made a sketch using an arbitrary atmospheric molecular backscatter coefficient profile and a height-constant particle backscatter coefficient (equal at both wavelengths) of 7e-6m^-1 sr^-1 in order to obtain a scattering ratio at 532 nm shortly above 5 as given by the authors as detection threshold for clouds. The input is illustrated below:

[Figure]

As one can see, the molecular backscatter coefficients vary significantly with altitude and between the wavelengths.

As a result, also the scattering ratio varies with altitude even though the particle backscatter coefficient is constant, as shown below:

[Figure]

This figures also shows that the simple empirical conversion applied by the authors is not valid and given the fact that pressure and temperature profiles are provided with Aeolus data, a much better job could have been done to account for the difference wavelengths of Aeolus and CALIOP.

Furthermore, assuming that the given threshold (SNR at 532 nm>5) is valid for detecting a cloud, it also indicates that a cloud with the same optical properties (same particle backscatter) would be detected differently at different heights. See Figure below (same as above but differ y-axis)

[Figure]

For the case I simulated, CALIOP would "detect" a cloud (orange line) for all heights while with the Aeolus converted SR profiles would not lead to a cloud detection (SNR below 5) for some parts of the height profile.

For the real, physical, conversion this is even true up to 12 km height (blue line).

Thus, using the scattering ratio for cloud detection is not appropriate in my opinion. You could have used the scattering ratio from the satellite data to calculate the real particle backscatter.

I.e. by using the molecular backscatter which you calculate from the meteorological data which is included in the satellite data(AUX-MET in Aeolus data).

Despite all my own doubts concerning this conversion, the authors themselves state: "We would like to stress here that no linear scaling applied uniformly to SRs at all heights could change the ratio of high cloud detection frequency to low cloud detection frequency of ALADIN."

Therefore, I wonder: Why they are doing so?

4. Scattering ratio threshold:

The choice of this threshold SR>5 is not clear to me and seems very arbitrary and without justification.

What happens if this threshold changes?

5. Different vertical resolution

The different vertical resolution for Aeolus and Calipso is not sufficiently discussed.

6. Language

Language and phrasing need to be improved. It is hardly understandable and not well explained. Please use simple sentences.
Furthermore, "insider information of Aeolus" need to be explained otherwise it is not understandable for non-Aeolus experts.

7. Specific comments in addition to the pdf (examples)

- Some statements are either simply wrong or wrongly phrased, e.g.:
  "…is characterized by lower sensitivity **to high clouds** above ~7 km than CALIOP, that we explain by lower SNR for ALADIN at these heights that is due both to physical reasons (smaller backscatter at 355 nm)"
  Why should there be a smaller backscatter at 355 nm? This is in absolute contradiction to all my knowledge! The particle backscatter coefficient could be equal in clouds (Angström of 0), but the molecular backscatter coefficient is for sure higher (see plots) and thus the total backscatter is for sure also higher! Could you please comment?

- Abstract: Just one of many examples:
  "(b) the cloud detection agreement is better for the lower layers. Above ~7 km, the ALADIN product demonstrates lower sensitivity because of lower backscatter at 355 nm"
  I do not understand this statement. First of all: What do you mean? The volume backscatter coefficient, the particle backscatter coefficient, the molecular backscatter coefficient? It is not clear! And I also do not know why any of these should be lower at 355 nm compared to 532 nm (and 1064 nm).

- Abstract last sentence: Is not understandable. What values are this? What is a cloud detection agreement value? Abstracts should be self-explaining and understandable.

- Not all references are in alphabetical order

- Some mistakes in the names of the references, please check

[revised manuscript text omitted]

Author:    Subject: Sticky Note        Date: 14 06 2021 11:11:55
Didn't you apply  SR>5? So why are SR below 5 are shown?

[Figure]

**Figure A1:** Correlation between individual pairs of CALIOP and ALADIN scattering ratio profiles, for all altitudes. The colors of the bins represent the occurrence frequencies for $0.2 \times 0.07$ SR bins, as a function of both CALIOP's $SR_{532}$ and ALADIN's $SR_{355}$. For each point along the diagonal, a Gaussian was fitted to the data points lying along a perpendicular transect and the central point of the Gaussian is plotted as a red filled circle. The white dashed line represents a linear fit to these points. For comparison, black 615 dashed line shows the fit given by Eq. 1.

---

## Referee Comment (RC3)

The paper presents and discussed the comparison of the scattering ratio products retrieved from ALADIN and CALIOP observations. The paper is interesting and falls within the skopes of the AMT. The manuscript is well structured and well written in the majority of it's extent. I would suggest the publication of this work after the consideration from the authors to revise the manuscript based on the following comments/suggestions, targeted to improve the clarity of their results.

**General comment:**

The authors should state clearly in the title that this study is dedicated to cloud products only.

The study should include a quantification to some extent, and discussion, on the percentage of the clouds not detected from the 2 lidars with the methodology used. Additionally a discussion is needed on the effect of these cloud-miss-detections on the results of the intercomparison per altitude (low, mid, high-level clouds).

Although the title clearly states that this is a comparison of the scattering ratio products retrieved from the 2 systems, in the discussion throughout the paper the authors comments are attributed to the 2 systems only. It should be more clear that different approaches for cloud detection products from the 2 missions could lead to different results. See also specific comment below.

**Specific comments**

**Page 1, line 22:** "the ALADIN product demonstrates lower sensitivity because of lower backscatter at 355 nm": This statement is not clear. The backscatter at 355 nm is not expected to be lower than at 532nm. Please explain and revise accordingly.

**Page 2, line 43**: "Despite an excellent daily coverage and daytime/nighttime observation capability (Menzel et al., 2016; Stubenrauch et al., 2017), the height uncertainty of the cloud products retrieved from the observations performed by these spaceborne instruments is limited by the width of their channels' contribution functions, which is on the order of hundreds of meters, and the vertical profile of the cloud cannot be retrieved with accuracy needed for climate feedback analysis." The sentence is confusing. Consider revising to make it easier to follow. Possible suggestion: "...is limited by the width of their channels' contribution functions (which is on the order of hundreds of meters), and their uncapability to retrieve the vertical profile of the cloud with accuracy needed for climate feedback analysis.

**Page 2, line 47:** "This drawback is eliminated by active sounders, the very nature of which is based on altitude-resolved detection of backscattered radiation, and the vertical profiles of the cloud parameters are available from the CALIOP (Cloud-Aerosol Lidar with Orthogonal Polarization)lidar (Winker et al., 2003)and CloudSat radar (Stephens et al., 2002) since 2006, CATS (Cloud-Aerosol Transport System) lidar on-board ISS provided measurements for over 33 months starting from the beginning of 2015(McGill et al., 2015).": Too big sentence, difficult to read. Consider revising.

**Page 4, line 106**: "In Fig.1(a-c), we show the observation geometry and sampling of ALADIN's L2A product as well as three variables retrieved from its observations..": consider revising as: "...as three simulated variables that can be retrieved from its observations..".

Page 4, line 110: "..the horizontal variability of the observed scene is nearly the same in latitudinal and longitudinal directions at 100 km distance": This is only valid for homogeneous scenes. Please revise accordingly.

**Page 4, line 120:** "The cloud variability along the satellite's track has been estimated from the gridded EAMv1 data using the parameterization of (Boutle et al., 2014). Figure1 also serves as an illustration to theoretically achievable cloud detection agreement discussed below.": Although the cloud variability is estimated, in the plot the scene is cloud free. As the paper mainly investigates clouds, it would be interesting to have a cloudy demonstration also in addition to Figure 1.

Page 4, line 123: "...scattering ratio (SR)..": Please write how the scattering ratio is calculated.

**Page 4, line 124**: "An important companion of such a column is a corresponding quality flag column,..... which can be then compared with that of CALIOP.": The description is vague, please write more clearly what filtering you used in the data.

**Page 5, line 141**: "Since the CALIOP is not a HSRL, the detailed information on AMB and APB is not available, and one has to compare the SR products.": One could also use the temperature and pressure profiles from NWP (provided with Aeolus & CALIPSO) to produce the particulate backscatter coefficient, and convert/compare these parameters. So this part should be revised to highlight the choice of this study and not state it as the only option.

**Page 5, line 145-150**: "The choice of the fitting parameter is not crucial for the purposes of the present work ... collocated data.": I strongly advise the authors to follow the comment of the first reviewer regarding the wavelength conversions. Alternatively, if they decide to keep the analysis as is, then please provide a detailed discussion on the uncertainties induced from this simplified conversion.

**Page 6, line 167**: "To avoid the risks associated with the solar contamination, we picked up only the night-time cases": As Aeolus is in dusk-dawn, still variability is expected in the PBL with the CALIPSO nighttime observations above land. Can you comment on that in the manuscript?

**Page 6, line 172:** "...we have performed a numerical experiment using the same calculated data as we used in Fig.1": Shouldn't they be stated as "simulations"?

**Page 6, line 173 – 180**: "This time... the passive observations": It is very hard to follow the approach. A scheme/flowchart would be useful.

**Page 6, line 182**: "Overall, we considered about 1E5 pairs of pseudo-collocated data and we present the results of cloud detection in Fig.3": Please include also the region and season(s) used to produce these pseudo-collocated data, which represent the outputs of Figure 3.

**Page 6, line 184:** "or each altitude bin, the cloud detection agreement is a ratio of a number of cases when both instruments have detected a cloud (SR>5) ….": Please elaborate this choice of cloud cut off (e.g. literature) and comment on the uncertainties on the cloud detection

induced from this choice for different altitudes. Could you include in results (Figure 3) and discuss, the percentage of the clouds missed to be detected, from the 2 sensors in your simulation, with the presented methodology?

**Page 7, section 3.1.** It should be stated clearly in the section that the discussion refers to the RS retrieved products used in this study from the 2 sensors. As for example, a study with the cloud statistics from the Atlid L2A and CALIPSO L2 backscatter coefficient product products may provide different results.

**Page 8, line 224**: "In Appendix A, we demonstrate the correlation between individual pairs of CALIOP and ALADIN SR profiles; the conclusion of this exercise is that it justifies using Eq.1, but the uncertainties of the analysis do not allow to refine the conversion coefficients". This statement is very strong. One could refine the conversion coefficients, independently of the uncertainties of the analysis. I support that the authors should formulate this statement to correctly reflect the choices and limitations.

**Page 8, line 229**: "This observation gives a hint that the instrumental part provides the backscatter information sufficient for some cloud detection up to 20km, but the detection algorithm suppresses noisy solutions." This sentence is not clear. Please improve the phrasing.

**Page 8, line 246**: "Below, we will also discuss the YES\_YES statistics normalized to cloud amount, but at this point we also want to study the other cases, which cannot be normalized this way" Consider to improve the phrasing.

**Page 9, line 283**: "This exercise is not aimed at revealing any altitude offset in backscatter signal registration, because this part of experimental setup is robust in both instruments". Consider improving the phrasing.

**Page 9, line 10:** "For each local peak found, we have searched for a peak or for a maximal value of CALIOP's SR profile in the vicinity of  $\pm 3$ km from the peak height determined from ALADIN". Consider including the information that only the 82% of the clouds are used for this comparison (according to the statistics presented in line 296-297.

**Page 9, line 304:** "As for the clouds between ~3km and ~10km height, the height sensitivity effects skew the effective cloud height detected by ALADIN downwardsby 0.5–1.0km", It is not clear which are the high sensitivity effects between 3 to 10 km. Maybe the authors could summarize them in a sentence again here. Also, please comment to what extent could the actual 100-km-cloud-variability at these altitudes be responsible for the deviation in the altitudes seen by Aladin and Caliop in these altitudes. It is not clear if the authors point out on the Aeolus capability to detect the top of the cloud, on the SR methodology capability for the same, or on the effect of the natural variability between the 2 instruments on their products.

**Figure 1:** "...ALADIN's observation paths for centers of averaged profiles ...": How they are averaged? In Aladin L2A resolution?

**Figure 1:** "This inclination is schematically shown as an inclined line lying in lidar curtain plane whereas the real projection to the same plane should be a vertical line": This part is hard to understand. Same comment for the part inside the manuscript.

Figure 2: Can the authors comment on the absence of collocated points between 0-60° lon at  $\Delta$ time < 6hrs?

**Figure 7**: No data is difficult to be distinguished from the -2km color, both have dark purple. Consider changing the no data color.

**Figure 9**: Consider adding the colorbar here also in the upper panel. Additionally, consider stating what the error bars account for.

**Figure A1**: The red points are not scaled in the same frequency ranges as the occurrence frequencies. Wouldn't that be better?

**Technical corrections:**

Page 4, line 101: "According to Flamant et al. (2017)."
Page 6, line 182: "Ansmann et al. (2007)"
Page 7, line 195: "...between the two products.."
Page 7, line 200: "..for the thw instruments"
Page 7, line 203: "Analyzing the Fig. 4"
Page 8, line 242: consider rephrasing to "from the sensitivity study.."
Page 8, line 237: consider rephrasing to "..behavior of the SR cloud detection product agreement.."
Figure 3: "...to the total number of simulations .."
Figure 7: "...+-3km vertical vicinity..."

---

## Author Response (AR1)

We thank Reviewer #1 for his/her analysis and comments on the paper. The responses to major and minor comments are given below. We marked the reviewer's and the author's comments by "RC:" and "AC:", respectively.

**General comments**

First of all, we want to admit that a simplistic conversion of scattering ratios provided in the first version of the manuscript appeared to be a source of confusion for the reviewers and we apologize for this. Moreover, the reviews helped us to recall that there are two definitions of scattering ratio itself and even though they both are aimed at estimating the contributions of particulate and molecular components to the backscattered radiation, they are not the same. In the present version, we added a section with all necessary definitions and conversion formulae. This section also appears to be helpful in the discussion of the potential discrepancy sources. The collocated dataset has been reprocessed and the new scattering ratios at 532nm have been calculated and analyzed. Despite changes in wavelength conversion methodology, the results and conclusions changed little. But, we noted a certain improvement of the overall agreement between the ALADIN and CALIPSO datasets (e.g. see the numbers representing the normalized cloud detection agreement at different heights).

**Major comments**

RC: The title does not reflect the content of the paper. In fact, the authors focus only on the cloud detection capability based on scattering ratios.

AC: The present version of the article puts more stress on the scattering ratios profiles. In addition, we updated the title to "Comparison of scattering ratio profiles retrieved from ALADIN/Aeolus and CALIOP/CALIPSO observations and preliminary estimates of cloud fraction profiles"

RC: Furthermore, the whole instruction deals only with clouds and not a single word about scattering ratios is written

AC: We now have a whole new section dedicated to definitions, including those of scattering ratios

RC: The scattering ratio which is the essential part of this manuscript has never been properly defined. According to the reference which is given, I assume that, "the ratio between the total backscatter by particles and molecules and the molecular backscatter" (according to Flamant, 2008) is meant, i.e. the ratio between the total backscatter (represented by particles and molecules) to the molecular backscatter.

AC: We agree that the scattering ratio was not properly defined in the previous version. Please, see the general comments above. Indeed, the quoted definition is what is used in ALADIN product, but a different definition is used in the literature for CALIPSO scattering ratio (as CALIPSO is not a HSRL lidar contrarily to ALADIN). A more sophisticated processing is needed than what was provided in the initial version of the manuscript, to convert the scattering ratio from ALADIN to a scattering ratio similar to CALIOP. We believe that this time both the definitions and the conversion are OK.

RC: The conversion the authors use to account for the different wavelengths of CALIOP and AEOLUS is poor. For example, I have made a sketch using an arbitrary atmospheric molecular backscatter coefficient profile and a height-constant particle backscatter coefficient (equal at both wavelengths) of 7e-6m^-1 sr^-1 in order to obtain a scattering ratio at 532 nm shortly above 5 as given by the authors as detection threshold for clouds

AC: First of all, we'd like to thank the Reviewer #1 for his/her efforts to estimate the SRs and the applicability of thresholds. Second, we were not using the same definition of SR as the reviewer in the previous version of the manuscript. Please, read the Section 3 of the present version of the manuscript, which should clarify SR definition, the wavelength conversion and the cloud detection threshold.

RC: Despite all my own doubts concerning this conversion, the authors themselves state: "We would like to stress here that no linear scaling applied uniformly to SRs at all heights could change the ratio of high cloud detection frequency to low cloud detection frequency of ALADIN." Therefore, I wonder: Why they are doing so?

AC: In the present version of the manuscript, we apply a proper conversion to SR'_532 and we discuss the potential sources of bias associated with the parameters of this conversion. We show that by adjusting the parameters of the conversion one can change the ratio between high- and low-level clouds, but there are physically defined limits for this "tweaking".

RC: The choice of this threshold SR>5 is not clear to me and seems very arbitrary and without justification.

AC: First of all, we draw the Reviewers' attention to the fact that the threshold is applied to "CALIOP-like" SR and not to "ALADIN-like" one (please, see Section 3 for the definitions). Second, the threshold SR>5 is used in CALIPSO-GOCCP product (Chepfer et al., 2008, 2013). It is derived from in depth analyses of the CALIPSO SNR in day time at vertical resolution 480m and horizontal resolution 330m, that has been defined within CFMIP for numerous scientific reasons. SR>5 is the threshold value that avoids false cloud detection in day-time due to low SNR induced by solar photons. Even though we used the nighttime cases for CALIOP, ALADIN's observations are in the twilight zone, so we decided to keep this threshold and to apply it uniformly to both instruments at all latitudes and heights.

RC: What happens if this threshold changes?

AC: The impact of this threshold change is discussed in (Chepfer et al. 2013) for CALIPSO. As for the present manuscript, we discussed the redistribution of the YES_YES, YES_NO and NO_YES cases with respect to threshold value in lines 269-274 of the previous version and we updated this discussion in Section 5.3 of the present version. Briefly, a uniform increase or decrease of the threshold for both SR products will not change the ratio between the ALADIN and CALIOP clouds because both will decrease or increase simultaneously. At the same time, a technical adjustment of the threshold for ALADIN's SR_532 could improve the agreement between the datasets, but there's a tradeoff between the YES_YES and NO_YES cases: by increasing the threshold we reduce the number of unexplained (see the text) NO_YES cases, but we reduce the number of good YES_YES cases. By lowering the threshold, we reduce the number of YES_NO cases, but we increase the number of NO_YES cases, a part of which is already difficult to explain. Nevertheless, the new plot with zonal cloud fractions (Fig. 7) looks promising.

RC: The different vertical resolution for Aeolus and Calipso is not sufficiently discussed

AC: In Section 3.1 and 3.2 of the present version that correspond to Sections 2.1 and 2.2 of the original one, we provide the information about the sampling of the instruments and about the resolution of the products used in collocation. Moreover, we apply the same cloud detection thresholds, on both SR(z)_CALIOP and SR(z)_ALADIN at the same vertical and horizontal resolutions.

RC: Language and phrasing need to be improved. It is hardly understandable and not well explained. Please use simple sentences.

AC: The text has been simplified and proof-read by a professional. We hope that this has improved the readability of the article.

RC: Furthermore, "insider information of Aeolus" need to be explained otherwise it is not understandable for non-Aeolus experts.

AC: We have removed internal variable names from the text and rewritten some explanations related to Aeolus in Section 4.5.

**Specific comments in addition to pdf**

RC: Some statements are either simply wrong or wrongly phrased, e.g.: "…is characterized by lower sensitivity to high clouds above ~7 km than CALIOP, that we explain by lower SNR for ALADIN at these heights that is due both to physical reasons (smaller backscatter at 355 nm)". Why should there be a smaller backscatter at 355 nm? This is in absolute contradiction to all my knowledge! The particle backscatter coefficient could be equal in clouds (Angström of 0), but the molecular backscatter coefficient is for sure higher (see plots) and thus the total backscatter is for sure also higher! Could you please comment?

AC: This statement is true and, indeed, the phrasing was misleading. We apologize for that. We meant the contribution of the particles to the total (particulate + molecular) signal. Even though the total backscatter is larger at 355nm, the particulate part can be buried in molecular return because the molecular backscatter is larger at 355nm while the backscatter from cloud particles is about the same. If the signal-to-noise ratio is small, then the cross-talk correction will be noisy and the particulate signal will be retrieved with large uncertainty. To avoid the confusion, in the present version of the manuscript we refer to the formalism defined in the second section and explain what we mean.

RC: Abstract: Just one of many examples: "(b) the cloud detection agreement is better for the lower layers. Above ~7 km, the ALADIN product demonstrates lower sensitivity because of lower backscatter at 355 nm" I do not understand this statement. First of all: What do you mean? The volume backscatter coefficient, the particle backscatter coefficient, the molecular backscatter coefficient? It is not clear! And I also do not know why any of these should be lower at 355 nm compared to 532 nm (and 1064 nm)

AC: We have rewritten the abstract for clarification.

RC: Abstract last sentence: Is not understandable. What values are this? What is a cloud detection agreement value? Abstracts should be self-explaining and understandable.

AC: Thank you for pointing this out. We have added the definition to the abstract. Please, see new Section 3.5 for the details.

RC: Not all references are in alphabetical order

AC: Fixed, thanks.

RC: Some mistakes in the names of the references, please check

AC: Fixed, thanks.

For the rest of the reviewer's comments in PDF, please, see below.

AC: some of the pages are cluttered with comments, so we could not add an answer beneath or near each of them. Instead, we provided the answers in the same order as they appear in the top of the page. Sometimes, as on this page, one answer covers several questions.

AC: we have rewritten the abstract and we introduced the normalized cloud detetion agreement, CDAnorm, in the Abstract

mments on amt-

Page: 1

Author:  Subject:  Sticky Note    Date: 14.06 2021 09:43:49
Abstract not understandable

Author:  Subject:  Comment on Text    Date: 14.06 2021 09:44:08
poor phrasing

Author:  Subject:  Comment on Text    Date: 14.06 2021 11:26:31
What do these numbers mean? It is not understanable without reading the paper
**Comparing scattering ratio products retrieved from ALADIN/Aeolus and CALIOP/CALIPSO observations: sensitivity, comparability, and temporal evolution**

Artem G. Feofilov[1], Hélène Chepfer[1], Vincent Noel[2], Rodrigo Guzman[1], Cyprien Gindre[1] and Marjolaine Chiriaco[3]

[1]LMD/IPSL, Sorbonne Université, UPMC Univ Paris 06, CNRS, École polytechnique, Palaiseau, 91128, France
[2]Laboratoire d'Aérologie, CNRS/UPS, Observatoire Midi-Pyrénées, 14 avenue Edouard Belin, Toulouse, France
[3]LATMOS/IPSL, Univ Versailles Saint-Quentin en Yvelines, Guyancourt, France

*Correspondence to:* Artem G. Feofilov (artem.feofilov@lmd.polytechnique.fr)

**Abstract.**

The spaceborne active sounders have been contributing invaluable vertically resolved information of atmospheric optical properties since the launch of CALIPSO (Cloud-Aerosol Lidar and Infrared Pathfinder Satellite Observation) in 2006. To ensure the continuity of climate studies and monitoring the global changes, one has to understand the differences between lidars operating at different wavelengths, flying at different orbits, and utilizing different observation geometries, receiving paths, and detectors. In this article, we show the results of an intercomparison study of ALADIN (Atmospheric Laser Doppler Instrument) and CALIOP (Cloud-Aerosol Lidar with Orthogonal Polarization) lidars using their scattering ratio (SR) products for the period of 28/06/2019–31/12/2019. We suggest an optimal set of collocation criteria ($\Delta$dist < 1°, $\Delta$time < 6h), which would give a representative set of collocated profiles and we show that for such a pair of instruments the theoretically achievable cloud detection agreement for the data collocated with aforementioned criteria is $0.77\pm0.17$. The analysis of a collocated database consisting of ~78000 pairs of collocated nighttime SR profiles revealed the following: (a) in the cloud-free area, the agreement is good indicating low frequency of false positive cloud detections by both instruments; (b) the cloud detection agreement is better for the lower layers. Above ~7 km, the ALADIN product demonstrates lower sensitivity because of lower backscatter at 355 nm and because of lower signal-to-noise ratio; (c) in 50% of the analyzed cases when ALADIN reported a low cloud not detected by CALIOP, the middle level cloud hindered the observations and perturbed the ALADIN's retrieval indicating the need for quality flag refining for such scenarios; (d) large sensitivity to lower clouds leads to skewing the ALADIN's cloud peaks down by ~0.5±0.4 km, but this effect does not alter any anomaly for the considered period, indicating that hot pixels temporal evolution of cloud agreement quality does not reveal any anomaly for the considered period, indicating that hot pixels and laser degradation effects in ALADIN have been mitigated at least down to the uncertainties in the following cloud detection agreement values: $61\pm16\%$, $34\pm18\%$, $24\pm10\%$, $26\pm10\%$, and $22\pm12\%$ at $0.75$ km, $2.25$ km, $6.75$ km, $8.75$ km, and $10.25$ km, respectively.

Author:    Subject: Comment on Text    Date: 14.06.2021 09:44:37
"f" missing

AC: fixed, thanks

[revised manuscript text omitted]

[Figure]
* * *
Page: 3

Author: Subject: Cross-Out Date: 17.05.2021 14:55:03

Author: Subject: Comment on Text Date: 14.06.2021 09:45:08
This statement is very vague

Author: Subject: Sticky Note Date: 17.05.2021 14:56:29
not a single word of scattering ratios so far

Author: Subject: Highlight Date: 17.05.2021 14:59:05

AC: we have rewritten and reorganized the text and we added a whole new section with definitions (Section 3). As for the phrase with "set tte stage", we have rewritten it to "In addition, the methods developed in this study and its conclusions will set the stage for the future comparison of the ATLID/EarthCare observations with other space-borne lidar". We cannot be more specific at this time.

AC: In the updated version of the manuscript, Section 3 is dedicated to the deifinitions and the SR conversion approach

AC: we did not get, why the PRF of 50Hz is marked.

[Figure]

(Bley et al., 2021), and for the majority of the period considered in this work (see below), the vertical sampling of both Mie
95 and Rayleigh channels between 2 km and 22 km was equal to 1 km whereas the sampling below 2 km varied from 0.25 to
1 km. The native horizontal resolution of 140 m of the instrument is sacrificed to achieve a better signal to noise ratio both
onboard by accumulating the detected profiles and on the ground by averaging the downloaded profiles at different steps of
the processing chain (Flamant et al., 2017)

The present study has been done using the pilot L2A dataset from Aeolus, Prototype v3.10, which is available for a limited
100 period of ALADIN's observations, from 28/06/2019 through the 31/12/2019. According to (Flamant et al., 2017), the L2A
data is produced from the L1B product of this instrument and it contains height profiles of Mie and Rayleigh co-polarized
backscatter and extinction coefficients, scattering ratios, and lidar ratios (Flamant et al., 2008; Lolli et al., 2013) along the right
line-of-sight. For the end user, the profiles are provided both per observation scale (87 km averages) and on smaller scales after
applying scene classification, but for the purposes of the present work the scattering ratio on the scale of 87 km is an optimal
105 choice.

In Fig. 1(a-c), we show the observation geometry and sampling of ALADIN's L2A product as well as three variables retrieved
from its observations, namely, the APB (Attenuated Particular Backscatter), the AMB (Attenuated Molecular Backscatter),
and the ATB (Attenuated Total Backscatter). The white dashed lines in Fig. 1 represent the line of sight of the instrument.
One has to note, however, that in the real life the ALADIN's line of sight is pointed perpendicular to the flight direction; at the
110 same time, the horizontal variability of the observed scene is nearly the same rate signal and longitudinal directions at
100 km distance, so the sketch gives an idea of the comparability of the physical parameters observed by ALADIN (Fig 1a-
c) and CALIOP (Fig 1d). The atmospheric scene used in Fig 1 has been calculated for demonstration purposes for two
wavelengths, 355 nm (Fig 1a,b,c) and 532 nm (Fig 1d) from the output of the EAMv1 (Energy Exascale Earth System Model
(E3SM)) atmospheric model version 1) atmospheric model (Rasch et al., 2019) for the conditions of autumn equinox in Northern
115 hemisphere. This data has been obtained with the help of the COSP2 (the Cloud Feedback Model Intercomparison Project
Observational Simulator Package, v2) package, which is capable of simulating the atmospheric observables for spaceborne
instruments (Swales et al., 2018) The CALIOP is built into COSP2 (hepfer et al., 2008) whereas the ALADIN is not yet a
part of this package, so we used the 355 nm calculations by COSP2 (Reverdy et al., 2015) at fine grid corresponding to
ALADIN's original laser pulse frequency rate and modified them in accordance with the ALADIN's vertical and horizontal
120 averaging. The cloud variability along the satellite's track has been estimated from the gridded EAMv1 data using the
parameterization of (Boutle et al., 2014). Figure 1 also serves as an illustration to theoretically achievable cloud detection
agreement discussed below

For each profile corresponding to an inclined dashed line in Fig. 1, we extracted the corresponding scattering ratio (SR) column
of tca_optical_properties group of variables where SCA stands for standard correct algorithm (Flamant et al., 2017). An
125 important companion of such a column is a corresponding quality flag column, which we scanned looking for the points
characterized either by high Mie signal-to-noise ratio (SNR) or by high Rayleigh SNR, and by a flag that indicates an absence

Page: 4

Author: Subject: Highlight Date: 14.06.2021 11:05:32
No reference given. You need to explain what this means and what is the difference to operational data

Author: Subject: Comment on Text Date: 18.05.2021 09:52:16
the scattering ratio you use is never defined, as it is essential for this work, you should do so

Author: Subject: Comment on Text Date: 17.05.2021 15:05:39
this is not true and heavily depends on the scene

Author: Subject: Comment on Text Date: 17.05.2021 15:07:53
it gives a wrong idea, because scenes are usually by far not so homogenous

Author: Subject: Comment on Text Date: 17.05.2021 15:08:41
for what do you need the autumn equinox in this simulatiuon?

Author: Subject: Comment on Text Date: 17.05.2021 15:09:42
what does this mean?

Author: Subject: Comment on Text Date: 14.06.2021 09:41:29
I don't understand, more explanation needed. I do not see any cloud in this figure

Author: Subject: Comment on Text Date: 14.06.2021 11:07:13
nobody who is not familiar with Aeolus L2A data structure will understand this

Furthermore, it is not trace-able/understandable for anyone not within an Aeolus Cal/Val team yet. A proper reference should be given or, if not available, a more detailed description of the data set needs to be given here. E.g. what is the difference to other Aeolus data. Why have you used this data set, etc

Author: Subject: Comment on Text Date: 14.06.2021 11:06:50
what does this mean: HIGH SNR? >2, >10, >100?

AC: In the new version, we write "the L2A product of this instrument and it contains height profiles of Mie and Rayleigh co-polarized backscatter and extinction coefficients, scattering ratios (SR), and lidar ratios (Flamant et al., 2017; Lolli et al., 2013) along the lidar line-of-sight". There's no reference per se, this is a test product and we have a corresponding statement in the Disclaimer at the end of the manuscript

AC: Please, see new Section 3

AC: we have updated the description of our numerical experiment

AC: Fig. 1 is now different

AC: There's nothing special about the Autumn equinox itself, this season just happens to be in the middle of the Prototype v3.10 data period.

AC: We got rid of internal Aeolus variable names in the present version of the manuscript

AC: the exact values of SNR used in the Aeolus algorithms are not given in the ATBD, so we just used the binary (yes/no) flags relying on the experience of the processing team.

Author: Subject: Comment on Text  Date: 14.06.2021 11:07:30
I am not sure if these flags are valid in these kind of data. These data are all preliminary. You should discuss this

Author: Subject: Comment on Text  Date: 18.05.2021 09:56:56
This statement is in contradiction to Figure 1, where you clearly see that it is not nadir but only close to nadir

Author: Subject: Comment on Text  Date: 18.05.2021 09:56:24
I guess you mean the altitude of the orbit, but this is not written here. Please state correctly

Author: Subject: Comment on Text  Date: 18.05.2021 10:26:48
GCM never explained

Author: Subject: Comment on Text  Date: 14.06.2021 09:54:36
This is not state of the art and not acceptable - see plots in my text

Author: Subject: Comment on Text  Date: 14.06.2021 09:55:42
I do not understand this statement. And the justification given in the Appendix is not sufficient in my opinion

AC: the Prototype version of the Aeolus data is supposed to be self-consistent. We have a Disclaimer at the end of the manuscript, which states that all the data in this version are preliminary.

AC: The small offset from nadir in CALIOP was introduced to reduce the surface reflection effects. This modification barely changed the optical path lengths, so it still can be called a "nadir-viewing instrument". To be precise, we changed it to "near nadir viewing lidar"

AC: We modified the phrasing about the orbital height, thaks for pointing this out

AC: GCM is now introduced in the explanation of the first abbreviation

AC: We agree that SR recalculation was oversimplified. In the present version of the manuscript, we have a whole new section dedicated to the definitions and recalculation approach.

AC: The validation part and its discussion have been removed
* * *
[Figure]

[revised manuscript text omitted]

* * *
Page: 6

**Author:** Subject: Comment on Text   Date: 14.06.2021 11:07:47
As Aeolus is flying at a dusk-dawn orbit, how can you select night time cases? Or is this valid for Calipso only? But then you have a bias, right?

**Author:** Subject: Comment on Text   Date: 14.06.2021 09:56:04
I really did not understand what you are doing here. Maybe a sketch or flowchart could help

Why do you need to imitate diurnal variation? Please explain!

**Author:** Subject: Comment on Text   Date: 14.06.2021 11:09:06
The choice of this threshold is not clear to me and seems very arbitrary. Furthermore, the use of a scattering ratio for cloud detection in questionable to me at all, as the scattering ratio depends on temperature and pressure and thus on height even with uniformly distributed particle load as shown in the attached plots. Using one threshold would mean that you detect a cloud at one height while you may not detect a similar cloud at another height. Thus, using the scattering ratio for cloud detection is not appropriate in my opinion. You could have used the scattering ratio from the satellite data to calculate the real particle backscatter coeff by using the molecular backscatter coeff which you calculate from the meteorological data which is included in the satellite data as well

**Author:** Subject: Comment on Text   Date: 18.05.2021 15:10:14
I do not "see" that. At least in Fig 3 it is not obvious. To what you are referring to? And can you give more explanation?

**Author:** Subject: Comment on Text   Date: 14.06.2021 10:00:39
Any evidence or proof for such a statement? Again to what are you referring?

**Author:** Subject: Comment on Text   Date: 18.05.2021 15:12:08
is this valid for ALADIN as well?

AC: Indeed, dusk-dawn observations are not equal to night-time observations, but this selection itself does not lead to a bias. We discuss the diurnal cycle effects in the manuscript and according to our estimates performed without local time filtering, the results are nearly the same. The idea here was to get rid of solar photons and to apply the same SR threshold for both intruments.

AC: A flowchart has been added and the text was updated

AC: As for the choice of the SR threshold, please, see the comments in the text version of the review.

AC: If one looks at Fig. 3 (now Fig. 4), one will see that the curves marked "w/o noise" and "w/noise" are virtually the same. The curves with noise correspond to variability caused by diurnal variation and instrumental noise added to the calculations. Therefore, the primary source of deviation from 1 is the observation geometry and the collocation quality.

AC: Saturation effects do not depend on the instrument

[Figure]

**Atmospheric Measurement Techniques Discussions**

Fig 3 for the sake of clarity) Overall, the theoretically achievable agreement for the collocated data at a given setup can be estimated as $0.77\pm0.17$ for cloud detection

**3 Results and discussion**

**3.1 Zonal averages**

195  To give a general overview of the agreement between two products, we have split the database to latitudinal zones: 90S–60S, 60S–30S, 30S–30N, 30N–60N, 60N–90N (Fig 4). As it was stated above, we recalc the $SR_{532}$ values retrieved from ALADIN observations to $SR_{532}$ using Eq 1. Even though the zonal mean statistics does not imply using collocated data, we do it to avoid any incoherence in sampling different geographic areas. By using exactly the same number of profiles collocated within 1°, we ensure the same coverage and sampling by both lidars. If the detection efficiency of different cloud types were the same

200  for two instruments, the plots would have been close to each other because the horizontal variability of clouds would cancel out due to averaging over a large number of profiles within the zone and the diurnal variation is small over oceans, which constitute two thirds of the cases used to build Fig 4 (Noel et al., 2018; Chepfer et al., 2019; Feofilov and Stubenrauch, 2019) Analyzing the Fig 4, one can note the following: (1) the SR altitude histograms of CALIOP (Fig 4a-e) are characterized by two distinct peaks corresponding to low-level and high-level clouds; this feature is coherent with other observations, e.g. with

205  GEWEX (Global Energy and Water cycle Experiment) cloud assessment (Stubenrauch et al., 2013); (2) the SR/altitude histograms built for SRs achieved from ALADIN's observations (Fig 4f-j) are characterized by a smoother occurrence frequency plot where the two-peak structure is less pronounced than for CALIOP; (3) even though ALADIN detects some clouds in polar stratosphere (PSCs), its overall sensitivity to high clouds (>7 km) is lower than that of CALIOP; (4) both rows show certain consistency of zone-to-zone change up to ~3km altitude while the behavior above requires a more detailed view

210  We would like to stress here that no linear scaling applied uniformly to SRs at all heights could change the ratio of high cloud detection frequency to low cloud detection frequency of ALADIN the same is true for CALIOP. In the next step, we compare the "instantaneous" profiles provided by CALIOP and ALADIN having in mind the peculiarities of cloud detection sensitivity differences observed in Fig 4

**3.2 Comparing pseudo-individual profiles at ALADIN's L2A product resolution**

215  To address the high cloud detection sensitivity, we have inspected the 6h nighttime subset of collocated data, looking for the cases, which would satisfy the following criteria: (1) both instruments should have at least one strong SR peak; (2) the vertical position of this peak detected by one instrument should match that of the peak detected by a second instrument within 1 km; (3) the CALIOP SR profile should have a secondary peak at or above 9 km (Fig 5a-j). For the comparison purposes, the panels in Fig 5 represent the individual profiles belonging to the same 5 zones as the panels of Fig 4. For the sake of simplicity, we

220  compare the $SR_{532}(z)$ profiles recalculated to $SR_{355}(z)$, but we also show the source $SR_{532}(z)$ profiles for reference purposes. Regarding the conversion using Eq 1, the strong peaks selected this way demonstrate a qualitative agreement between the
* * *
**Comments:**

Author:  Subject Comment on Text  Date: 18 05 2021 15:14:57
which products? SR or cloud product? Unclear

Author:  Subject Comment on Text  Date: 18 05 2021 15:15:04
phrasing

Author:  Subject Comment on Text  Date: 18 05 2021 15:16:04
phrasing

Author:  Subject Comment on Text  Date: 18 05 2021 15:17:14
maybe because your transformation of the SR is incorrect?

Author:  Subject Comment on Text  Date: 18 05 2021 15:18:07
If you know that, why have you done so?

Author:  Subject Comment on Text  Date: 14 06 2021 09:07:46
What is true for CALIOP?

Author:  Subject Comment on Text  Date: 14 06 2021 11:09:47
is the feasible according to the range-bin setting of Aeolus. In principle, Aeolus can also have 2 km thick range bins. Can you comment on this?

Author:  Subject Comment on Text  Date: 14 06 2021 10:08:26
I do not see a reason to use the "sake of simplicity" here. A correct conversion is needed

AC: now we specify that we compare SR(532nm,z) and SR'(532nm,z)

AC: we have changed the phrasing

AC: Indeed, the updated transformation of SR gives somewhat better results, but the general conclusions (and the one, which is marked by the Reviewer on this page) remain the same

AC: please, see our answer regarding linear conversion in the text portion of the replies above

AC: That's true, 2km range bins can exist in ALADIN data, but they were not the subject of a case study described here (and we did not see them). As for the averaged plot, they will not spoil the picture, either, because the data is interpolated to a regular grid and then averaged.

AC: We do not use SR355 anymore and we apply an updated (and presumably correct) conversion procedure.

Author: Subject: Comment on Text Date: 14.06 2021 10:09:39
This is not convincing

Author: Subject: Comment on Text Date: 14.06 2021 10:10:13
how do you account for the different vertical resolution?

Author: Subject: Comment on Text Date: 14.06 2021 10:11:10
This statement is not clear for me

Author: Subject: Comment on Text Date: 14.06 2021 10:14:44
Phrasing needs to be improved

AC: We do not have Appendix A and the corresponding discussion in the present version of the manuscript.

AC: When the profiles are compared, the resolution of CALIOP is already lowered through averaging.

AC: We've added an explanation after this phrase

AC: This section has beed rewritten

peak values calculated from $SR_{355}$ and peak retrieved $SR_{532}$ values. In Appendix A, we demonstrate the correlation between individual pairs of CALIOP and ALADIN SR profiles: the conclusion of this exercise is that it justifies using Eq. 1, but the uncertainties of the analysis do not allow to refine the conversion coefficients. As for the potential capability of ALADIN to

225   detect high clouds, the subset Fig. 5a-e represents the cases, for which the instrument was capable of retrieving the peak of the same magnitude and height as the peak detected by CALIOP. Even though these cases exist, they are far less frequent than those shown in Fig. 5f-j. We did not detect and correlation between the collocation criteria (Δdist; Δtime) and the frequency of occurrence of these cases, it's just a statistical observation that both types of cases exist and the former are less frequent than the latter. This observation gives a hint that the instrumental part provides the backscatter information sufficient for some

230   cloud detection up to 20 km, but the detection algorithm suppresses noisy solutions. The PSC detection discussed below (see also Fig. 4f) confirms this assumption because the vertical extent and the composition of these clouds yield a strong signal. Further speculations on this subject might be beyond the scope of the present article, but we believe that the high cloud detection agreement might be improved by studying the collocated cases provided in the supplementary materials and by applying different noise filtering techniques in the L0→L1→L2 elements of the ALADIN retrieval chain. Figures 5k-o will be discussed

235   below in the context of low-level cloud observations.

**3.3 Cloud detection agreement**

To illustrate the peculiarities of zonal and altitudinal behavior of cloud detection agreement between two considered instruments, we have split the collocated data into four groups (Fig. 6). For each altitude/latitude grid point, we have estimated the number of cases when both instruments have detected a cloud ($SR_{532}(z)>b$), when neither of instruments has detected a

240   cloud, when only CALIOP has detected a cloud, and when only ALADIN has detected a cloud. For the sake of simplicity, we will call them YES_YES, NO_NO, YES_NO, and NO_YES cases. It is clear that in the ideal experiment the number of mismatched cases (YES_NO and NO_YES) should tend to zero. From the study presented in Section 2.4, we expect that the ratio of (YES_YES+NO_NO)/(YES_YES+NO_NO+YES_NO+NO_YES) should be about 0.77±0.17 if both instruments detect the clouds with the same efficiency. In Fig. 6a we show the ratio of YES_YES cases to the total number of collocated

245   profiles per altitude/latitude bin. This panels resembles a typical cloud amount plot, and this is expected because in the case of an ideal agreement the aforementioned ratio is equivalent to cloud amount definition. Below, we will also discuss the YES_YES statistics normalized to cloud amount, but at this point we also want to study the other cases, which cannot be normalized this way. Even though the distribution in Fig. 6a looks physical, the absolute numbers are somewhat low and this is explained by YES_NO and NO_YES distributions (Fig. 6c and d, respectively) As for NO_NO agreement (Fig. 6b), it is

250   close to 100% in the high-altitude area where there are no clouds. This indicates that the noise-induced false detection rate of both instruments is low, and this is a good sign.

If we consider the mismatch of YES_NO type (Fig. 6c), we will see that the altitudinal/zonal distribution of the mismatch occurrence frequency resembles that of the YES_YES type. A part of mismatch can be explained by theoretically allowed cloud detection disagreement discussed in Section 2.4. However, the occurrence frequency of YES_NO cases above 3 km is

**Author:** Subject: Comment on Text    Date: 14.06.2021 10:17:13
What does it mean? It could be also a cause of your rough conversion of the scattering ratio and or the range-bin thickness of Aeolus?

**Author:** Subject: Comment on Text    Date: 14.06.2021 10:18:28
Phrasing! Is this really only one cloud?

**Author:** Subject: Comment on Text    Date: 14.06.2021 10:19:55
Can you explain, how these false peaks could develop? It is not clear to me

AC: as we wrote before, the updated conversion algorithm did not change the magnitude of SRs for high clouds. In any case, the agreement of the updated version is somewhat better, so we changed the phrasing.

AC: this time, we consider all possible reasons for NO_YES cases, including those related to recalculation procedure. Our conclusion is that even if we tweak the conversion parameters, we will explain only half of these cases

AC: We do not know the exact details of the algorithms, so we can only speculate here using the basics of active remote sensing. Since the lidar equation (Eq. 1) is solved layer per layer and the upper layers affect the solution for the lower one,s the "false peaks" we were speaking about, can appear if the solution in the upper layer is perturbed by noise. We have added the explanations and toned down the phrasing of this section.
* * *
[Figure]

255 roughly that of YES_YES cases, and this indicates the retrieval sensitivity issue of ALADIN. The NO_YES mismatches
(Fig 6d) require specific attention because they are not expected from the methodological point of view: the cloud extinction
at 355 nm is larger than at 532 nm and the observation geometry of ALADIN makes the optical paths $1/\cos(SVA) = 1.22$
times longer than those for CALIOP, where SVA stands for satellite viewing angle $\approx 35°$. The typical individual profiles
corresponding to NO_YES mismatches are shown in Fig. 5k-o. As one can see, despite the unfavorable observation conditions

260 (e.g. an opaque cloud with peak $SR_{532}$ value of ~22 at 9 km in Fig 5.l), ALADIN reports two valid points beneath the cloud
whereas it does not report anything at 9 km height where CALIOP sees a thick cloud. These cases do need our special attention.
On the one hand, many cases of this type are over the ocean, so one can rule out the surface echo mixed with atmospheric
backscatter and treated like an atmospheric signal. On the other hand, the NO_YES cases are often accompanied by the
structures similar to those presented in Fig. 5k,l,n which are probably provoked by a presence of a cloud at these heights. The

265 perturbations to the extinction and backscatter profile caused by these structures might propagate downwards, thus causing the
appearance of the false peaks in the lower layers of ALADIN's data. This indicates a need for a quality flag refinement in the
lower layers in the presence of a thick cloud above and the improvement of thick cloud detection itself. Apparently, the
CALIOP cloud retrievals beneath thick clouds do not suffer from these effects.
To test whether the aforementioned disagreements are at least partially caused by the cloud definition and SR recalculation to

270 another wavelength and whether the agreement could be improved, we varied the SR threshold for ALADIN, assuming no optimum value for
$\pm 50\%$ uncertainty on the parameters forming the coefficients of Eq. 1. However, this exercise yielded no optimum value for
SR threshold: its lowering for ALADIN increased the number of YES_NO cases and reduced the number of YES_NO cases, but
at the same time it increased the frequency of NO_YES cases. Correspondingly, increasing the threshold reduced the number
of NO_YES cases, but it adversely affected the YES_YES agreement. Summarizing this comparison, one can conclude that

275 (a) a cloud detected by CALIOP is detected by ALADIN in ~50% of cases for clouds below ~3km and in ~30% of cases for
higher clouds; (b) in the cloud-free area, the agreement between the datasets is good that indicates a low frequency of false
positive detections by both instruments; (c) one half of the cases when ALADIN detects a cloud missed by CALIOP should
be attributed to false positive detection of the low cloud in the presence of a higher opaque cloud, which perturbs the retrieval
in the lower layers.

280 **3.4 Cloud altitude detection sensitivity**

Besides marking the profile elements as "cloudy" and "not cloudy" and comparing the cloud detection statistics as we did in
the previous section, it would be interesting to obtain cloud peak detection statistics for pairs of collocated profiles like those
shown in Fig. 5. This exercise is not aimed at revealing any altitude offset in backscatter signal registration, because this part
of experimental setup is robust in both instruments. But, as we saw in Fig. 4 and Fig. 6, the sensitivity of ALADIN to high

285 clouds is lower than to lower clouds and a convolution of sensitivity curve with the backscatter profile can skew the cloud
peak position and the average cloud height. To illustrate this effect, we have carried out the following analysis. For each pair

Author:   Subject: Comment on Text   Date: 14.06.2021 10:22:48
Altitude?

Author:   Subject: Highlight   Date: 17.05.2021 15:19:17

AC: yes, we meant the altitude, thanks

AC: fixed, thanks

[revised manuscript text omitted]

Author: Subject: Comment on Text Date: 14.06.2021 10:28:59
why should there be a smaller backscatter at 355? This is in absolute contradiction to all my knowledge? Particle backscatter coefficient could be equal (Angström 0). Molecular backscatter coefficient is for sure higher and thus total backscatter is for sure also higher!

Author: Subject: Comment on Text Date: 14.06.2021 10:30:01
phrasing poor

AC: please, see our comment in the text section. What was meant was the "information content" of particulate backscatter with its noise with respect to molecular one, not the signal itself. Please, apologize for the confusion.

AC: we have rewritten this section

[revised manuscript text omitted]

Page: 15

Author: Subject: Highlight Date: 14.06.2021 11:28:48
This is no proper reference as not public available

AC: This is true, but this is the only source of information available. A comment from a Technical Editor is needed for such a case.

[revised manuscript text omitted]

Author:   Subject: Comment on Text     Date: 17 05 2021 15:20:45
250-2000

Author:   Subject: Comment on Text     Date: 17 05 2021 15:20:56
125-2000

AC: Fixed, thanks

| Instrument | Orbit inclination [deg] | Equator crossing LT [h] | Off-nadir angle [deg] | PRF [Hz] | Native resolution [m] | L2 resolution resolution [m] |
|---|---|---|---|---|---|---|
| ALADIN | 96 97 | 6:00 / 18:00 | 35 | 50.0 | 140 (H) x 1000 (V) | 87000 (H) x 1000 (V) |
| CALIOP | 98 00 | 01:30 / 13:30 | 3 | 20.1 | 333 (H) x 60 (V) | 333 (H) x 500 (V) |

Table 1: Comparison of orbital parameters, viewing geometries, and resolutions of ALADIN and CALIOP instruments

| Δtime [h] | Daytime ×1E3 | Night-time ×1E3 | Total ×1E3 | Remarks |
|---|---|---|---|---|
| < 1 | 4 3 | 3 7 | 8 | Narrow polar zone |
| < 3 | 13 1 | 11 2 | 24 3 | Broader polar zone |
| < 6 | 91 | 78 | 169 | All zones covered |
| < 12 | 135 | 116 | 251 | Unequal distribution of Δtime |
| < 24 | 176 | 146 | 322 | Unequal distribution of Δtime |

Table 2: Number of collocated cases for Δdist < 1° and different Δtime values

555

Author: Subject: Sticky Note    Date: 17 05 2021 15:06:59
only valid for a cloud free scene IF at all

Author: Subject: Sticky Note    Date: 14 06 2021 11:30:37
hard to understand  A clearer description is needed in the caption what the dashed lines mean  It is also not clearly evident while reading the
text  Furthermore, I recommend to use different colours for the dashed line and then clearly describe which dashed line represents what

AC: Fig. 1 has been replaced with a 3D orbital view. The explanation of the numerical experiment refers more to a flowchart in Fig. 3

[Figure]

Figure 1: Observation geometry, averaging, and retrieved parameters for (a–e) ALADIN/Aeolus at its L2A resolution of 87 km and (d) CALIOP/CALIPSO at its native resolution: (a) Attenuated particular backscatter (APB) at 355 nm; (b) Attenuated molecular backscatter (AMB) at 355 nm; (c) Attenuated total backscatter (ATB) at 355 nm; (d) Attenuated total backscatter (ATB) at 532 nm. The scene has been calculated for demonstration purposes using COSP2 simulations with the EAMV1 model data as an input. White dashed lines stand in (a–c) for ALADIN's observation paths for centers of averaged profiles and in (d) for CALIOP averaged observation path corresponding to averaged ALADIN on the left and for individual CALIOP profiles on the right (with its 3° off-nadir viewing angle). ALADIN observes the atmosphere at 35° to the nadir and perpendicular to the flight direction. This inclination is schematically shown as an inclined line lying in lidar curtain plane whereas the real projection to the same plane should be a vertical line.

Author:    Subject: Sticky Note      Date: 18 05 2021 15:21:16
why there is a gap?

AC: this is a good question - due to large overhead at the collocation, we did not read the previous or next day. As a result, the collocation algorithm did not find anything for the data measured, for example, 4h earlier at a given longitude. In the present version, this figure has been replaced with 2D histograms in the latitudinal bins, but the gap remains in the collocated dataset.

[Figure]

Figure 2: Geographical distribution of collocated points for (a) Δtime < 1 h; (b) Δtime < 6 h; (c) Δtime < 24 h for Δdist < 1°.

Time difference [h] between ALADIN and CALIOP profiles colocated within 1°

-24   -18   -12   -6   0   6   12   18   24

Author: Subject: Comment on Text   Date: 18 05 2021 15:23:40
Not really understandable, please rephrase

AC: we have updated the figure caption

570

[Figure]

**Figure 3:** Estimating theoretical cloud detection agreement using pseudo-collocated scattering ratio (SR) data calculated using COSP2 lidar simulator coupled with the output of the EAMv1 atmospheric model. For each altitude bin, the agreement is defined as a ratio of number of cases when both CALIOP and ALADIN have detected a cloud to a total number of simulations for a given bin. The cloud amount corresponds to a ratio of number of cases when a reference CALIOP SR value without noise was above the detection threshold to a total number of simulations for a given bin. The normalized cloud detection agreement represents a ratio of the former to the latter.

575

Author: Subject: Sticky Note Date: 18 05 2021 15:25:42
you may consider using white backgrounds

AC: we tried white background, but it didn't improve the image. Instead, we zoomed in and moved the left-hand-side limit to SR=3 to show more of small SR values. We believe, this made the figure more informative.

[Figure]

580 **Figure 4:** Zonal mean comparison for the Δtime < 6h, Δdist < 1° collocated nighttime data subset (see Table 2): (a)-(e) CALIOP averages; (f)-(j) ALADIN averages, converted to SR at 532 nm for comparison purposes; (a,f) 90S-60S; (b,g) 60S-30S; (c,h) 30S-30N; (d,i) 30N-60N; (e,j) 60N-90N.

[Figure]

Figure 5: Pseudo-instantaneous comparisons of collocated ALADIN L2A SR profiles and CALIOP SR profiles averaged over 67 km along the track: (a, f, k) 90S-60S; (b, g, l) 60S-30S; (c, h, m) 30S-30N; (d, i, n) 30N-60N; (e, j, o) 60N-90N; (a-e) cases confirming ALADIN's capability to detect high-level clouds; (f-j) cases showing the cases when ALADIN misses a high cloud detected by CALIOP; (k-n) cases explaining the presumably false detection of a low level cloud by ALADIN; (o) a case with a real low cloud detected by both instruments with an extra point near the surface reported by ALADIN.

585

[Figure]

Figure 6: Cloud detection agreement: a) both CALIOP and ALADIN have detected a cloud (YES/YES cases); b) neither CALIOP nor ALADIN has detected a cloud (NO/NO cases); c) CALIOP has detected a cloud whereas ALADIN has not detected a cloud (YES/NO cases); d) CALIOP does not detect a cloud whereas ALADIN has detected a cloud (NO/YES cases).

[Figure]

595    **Figure 7: Cloud altitude detection sensitivity represented as a height difference between the CALIOP local peak height and corresponding ALADIN's cloud peak height or maximal SR height found in the ±3 km vicinity of CALIPSO peak. The subset corresponding to YES_YES selection (Fig. 6a) was used. White dashed isoline corresponds to colored area in Fig. 6a (occurrence frequency of about 5% and higher).**

[Figure]

**Figure 8: Temporal evolution of cloud detection agreement for the period of 28/06/2019-31/12/2019. The legend is consistent with that of Fig. 6: a) YES/YES; b) NO/NO; c) YES/NO; d) NO/YES. White vertical dashed lines correspond to the period of Air Motion Vector (AMV) campaign (28/10/2019–10/11/2019), which is characterized by smaller bin sizes and, therefore, larger SNRs for Mie and Rayleigh channels up to the height of 2250m.**

600

[Figure]

Figure 9: Temporal evolution of normalized cloud detection agreement for the period of 28/06/2019-31/12/2019: a) YES_YES statistics of Fig. 8a normalized by cloud amount; b) the same information presented for 5 heights as linear fits in 2D with error bars. The color scheme for panel (a) is consistent with that of Fig. 8.

605

Author: Subject: Sticky Note Date: 14.06.2021 11:11:55
Didn't you apply SR>5? So why are SR below 5 are shown?

AC: this Figure does not exist in the new version. As for the question, we wanted to check the conversion itself, regardless of the SR threshold used later.

[Figure]

Figure A1: Correlation between individual pairs of CALIOP and ALADIN scattering ratio profiles, for all altitudes. The colors of the bins represent the occurrence frequencies for 0.2 × 0.07 SR bins, as a function of both CALIOP's $SR_{532}$ and ALADIN's $SR_{355}$. For each point along the diagonal, a Gaussian was fitted to the data points lying along a perpendicular transect and the central point of the Gaussian is plotted as a red filled circle. The white dashed line represents a linear fit to these points. For comparison, black dashed line shows the fit given by Eq. 1.

We thank Reviewer #2 for his/her analysis and comments on the paper. The responses to major and minor comments are given below. We marked the reviewer's and the author's comments by "RC:" and "AC:", respectively.

**Major comments**

RC: These findings are quite valuable to understand how to interpret both data sets and also valuable to construct longer time records than those obtained by lidar on a single satellite. There is a lack of clarifications in the current form of manuscript.

AC: We thank the Reviewer for pointing out the importance of the work for merging the different space-borne datasets into one long-term record. As for the clarifications, we have added the definition of the Scattering Ratio, the formalism to convert the scattering ratio from 532 to 355nm, and the definition of the different variables (Sect. 3). We have also updated the figures and the corresponding text, and we have addressed all the comments of all the reviewers.

RC: Theoretical justification of using the simple SR conversion factor method between 355nm and 532nm in Equation (1) is not sufficient.

AC: We agree with this statement. We have added a section with all necessary definitions and conversion formulae. This section also appears to be helpful in the discussion of the potential sources of discrepancy between CALIPSO and ALADIN. The collocated dataset has been reprocessed and the conversion has been re-calculated and analyzed

RC: When model outputs are used, there is no need to rely on the conversion factor and the SR for 355nm and 532nm/1064nm can be estimated independently.

AC: This is true, but we do not used this conversion factor for the model+simulator part. We have re-written the simulation section, and we added a flowchart to clarify the steps of this simulation experiment.

RC: The choice in Equation (1) seems to be essential to the theoretical derived value (0.81) for cloud detection agreement between CALIOP and ALADIN. That is, the treatment of model output as well as cloud detection algorithm affect the estimation of the value of 0.81.

AC: Please, see the answer to the previous question. The theoretically estimate of the best achievable normalized cloud detection agreement (= value of 0.81, refined in this version) does not use Eq. 1. As we show in Fig. 4 of the new version of the manuscript, the value is mostly determined by difference in observation geometry and orbital parameters leading to non-ideal collocation.

RC: There are no descriptions about the output parameters for EAMv1 model used in this article.

AC: The outputs of the EAMv1 model are the usual standard inputs for COSP/lidar (e. g. Chepfer et al. 2008; Tang et al. 2019). But, we added several modifications to a standard model+COSP/lidar simulation for this study. Those are presented in the flowchart (Fig. 3) and described in Section 4: (a) subscale horizontal cloud variability; (b) instrumental noises for ALADIN and CALIOP; (c) diurnal variation of cloud fraction.

RC: The actual signals in the CALIOP and ALADIN contain the aerosols as well as clouds and molecules. Aerosol signals at 355nm might be larger than those at 532nm and it is naturally expected that the discrimination between clouds and aerosols is more challenging at ALADIN compared with CALIOP.

AC: First, we did not try to build the cloud detection scheme based on ALADIN-defined SR (see Eq. 2 in new version). As for the CALIOP-like defined SRs (new Fig. 5), the SRs from CALIOP are equal or larger than those estimated from ALADIN, so the cloud-aerosol discrimination problem mentioned in the question is not revealed.

RC: It is not clear how to incorporate the wavelength dependence of aerosols into the equation (1). It is not clear whether aerosols are contained in the EAMv1 model or not. There is no description about how multiple scattering effects for CALIOP and ALADIN are treated in the simulations in section 2.

AC: Again, the simulation experiment does not use Eq. 1. We apologize for a lack of clarity in the previous version of the manuscript regarding the simulations and we hope the new Sect. 4 is helpful. However, the question about multiple scattering is relevant and it is included into the present version of the manuscript in its new theoretical part (Sect. 2) as well as in the discussion of possible reasons for the discrepancy of low-level clouds.

RC: It seems to be possible to apply practically the same cloud detection algorithms used in the ALADIN L2A as well as CALIOP GOCCP products in the theoretical analyses in section 2. If one will do so, it would give a different cloud detection agreement of 0.77. The above-mentioned information is important to interpret the results in section 3 and conclusions.

AC: Since we did not convert the SRs for the simulation study (but only for the actual observations), we actually apply the same detection algorithms to the ALADIN an CALIPSO theoretical analyses. We agree that it was not well described in the previous of the manuscript, we hope the new Sect. 4 and the flowchart help.

RC: There are also lack of clarifications in the treatment of CALIOP clouds for the comparisons. It seems there is no sub-grid scale treatment for 87km-ALADIN L2A products so that 0 or 1 cloud fraction for each 87km-grid.

AC: First, the sub-grid treatment of ALADIN is a part of a Prototype v_3.10 algorithm from ESA, which is not available for the end user. The current end-user ALADIN dataset contains the backscatter and extinction profiles at 355nm that are standard for an HSRLidar (but not for non-HSRL like CALIOP). There's no 0 or 1 in this ALADIN dataset nor does it define the cloud fraction itself. Therefore, we performed a conversion from ALADIN's backscatter and extinction at 355 to SR'_532 and apply the uniformly defined cloud detection threshold on this SR'_532 profile (see Section 2 in the updated version of the manuscript). Second, we used high-resolution CALIOP data on 333m grid, averaged its AMB(z) and ATB(z) profiles at the same vertical and horizontal resolution as ALADIN and calculated SR_532(z). These procedures ensure that the two averaged profiles (SR'_532 derived from ALADIN and SR_532 derived from CALIOP) are comparable.

RC: On the other hands, CALIOP product has finer resolution (333m or 1km). It is not clear how to treat cloud fraction for CALIOP after 67km averaging for the comparisons compared with ALADIN in sections 2 and 3.

AC: We do not use the existing cloud fraction from CALIOP. As mentioned above, we averaged ATB and AMB(=ATBmol) over similar resolution as ALADIN and only then do compute SR and apply the cloud detection threshold. We are well aware of the fact that this might lead to an overestimation of cloud fraction in the boundary layer, but we perform this procedure to ensure the comparability of two datasets.

RC: Brief description of Aeolus L2A cloud product is also instructive.

AC: Such a product doesn't exist (yet), we defined the cloudy or non-cloudy bins by applying the cloud detection threshold to SR_532(z) values.

RC: The SR for CALIOP was originally estimated to create CALIPOSO GOCCP products where Equation (1) is not needed. It is not convincing why equation (1) is used to simulate SR at 532nm.

AC: In the present version of the manuscript, we do not use Eq. 1 anymore. Instead, we use a more precise recalculation approach presented in Section 3. But, the idea of converting ALADIN's 355 data to 532nm was to compare apples to apples and apply the same cloud detection threshold to the 'same' SR profile at the same spatial resolution.

RC: After reading the manuscript several times, any reasonable explanation was not found why the upper clouds are smaller for ALADIN compared with CALIOP, though CALIOP did not detect most of PSCs where ALADIN detected (in shown in the Figure 4a and f).

AC: Actually, we discussed PSC detection in lines 230-231, 301-303, and 374-376 of the previous version of the manuscript, but in the rest of the manuscript there was a confusing explanation regarding the particulate backscatter and we apologize for this. As we wrote in response to the Reviewer #1's question, we meant the detection of the particles. Even though the total backscatter is larger at 355nm, the particulate part can be buried in molecular return. If the signal-to-noise ratio is small, then the cross-talk correction (used in High Spectral Resolution lidar) will be noisy and the particulate signal will be retrieved with large uncertainty. We do not know the details of the L2 algorithm computing SR, extinction and backscatter used in ALADIN products, but a common sense tells us that if the signal is noisy then there's a high chance that the algorithm will reject it. Summarizing, our explanation of smaller ALADIN's sensitivity to high clouds is linked with a combination of weaker-than-planned SNR and smaller particulate backscatter compared to molecular one.

RC: The authors attributed the lower sensitivity of high clouds for ALADIN to smaller backscatter at 355nm without conducting further analysis.

AC: Please, see the previous answer for the corrected explanation. The text of the manuscript has been also updated to avoid misunderstanding.

RC: More discussion of the discrepancies in the cloud detections are requested. It is also noted that it is well established that CALIOP has a good capability to detect PSCs so that Figure 4a is strange.

AC: Please, check the new version of Fig. 4 (now Fig. 5) where we show the SRs starting from SR=3. In Fig. 5, one can also see the PSCs detected by CALIOP with SR>5. Note that this threshold is not optimized for PSC that can be optically thin. And, last, but not least, Fig. 8a does contain the PSCs, but their frequency of occurrence is low.

RC: There are several CALIOP based global cloud products, including NASA Langley's VFM products, GOCCP, DARDAR and KU cloud products and large differences were reported in (Cesana et al., 2016) JGR among GOCCP, NASA standard and KU products, indicating the different cloud detection methods caused the differences. There are several ways to bridge gaps between CALIOP and AEOLUS. Some comments are needed in this regard.

AC: The works mentioned by the reviewer are all using the same source that is L1 collected by CALIPSO. For comparing ALADIN and CALIPSO, the main challenges are because of the difference of nature of their L1 data: (1) ALADIN measures APB and AMB (and not ATB) because it is an HSRL, while CALIPSO measures ATB (and not APB and AMB) because it is a non-HSRL (See Eqs. in Sect. 3), (2) the wavelengths are different (355 nm vs 532nm), (3) the orbits and overpass times are different (see Sect. 2). We tried to state these points more clearly in the new version of the manuscript.

**Specific comments**

RC: p.6 line 182-184, need clarification for the methods and typical values of noises for Aeolus and CALIOP in the target data sets.

AC: We have updated the methodological part (see new Section 3). As for the noise values, we estimated them from the upper part of the vertical profiles, which are cloud-free and contain only molecular return, which is supposed to be smooth. We added this information to the manuscript (Section 4.1)

RC: p.25 Figure 6, zonal mean cloud frequency for CALIOP and ALADIN would be preferable prior to Figures 6a-d.

AC: Thank you for this suggestion. We added the requested figure and the corresponding text. It is interesting to note that visually the cloud distributions for the compared instruments are much more alike than the SR distributions. But, cloud detection threshold for higher clouds is reached less frequently for ALADIN than for CALIOP.

We thank Reviewer #3 for his/her analysis and comments on the paper. The responses to major and minor comments are given below. We marked the reviewer's and the author's comments by "RC:" and "AC:", respectively.

**Major comments**

RC: The authors should state clearly in the title that this study is dedicated to cloud products only.

AC: The present version of the article puts more stress on the absolute values of scattering ratios themselves. In addition, we updated the title to "Comparison of scattering ratio profiles retrieved from ALADIN/Aeolus and CALIOP/CALIPSO observations and preliminary estimates of cloud fraction profiles"

RC: The study should include a quantification to some extent, and discussion, on the percentage of the clouds not detected from the 2 lidars with the methodology used. Additionally, a discussion is needed on the effect of these cloud-miss-detections on the results of the intercomparison per altitude (low, mid, high-level clouds).

AC: If we understand this question correctly, it is related to the evaluation of clouds in the GCMs, and this question has been already addressed in (Chepher et al., 2008). For the current work, we are looking for similarities/differences in scattering ratio and cloud fraction profiles between the two lidar missions. If some clouds are filtered out in our approach, they are filtered out in the same way for both lidars.

RC: Although the title clearly states that this is a comparison of the scattering ratio products retrieved from the 2 systems, in the discussion throughout the paper the authors comments are attributed to the 2 systems only. It should be more clear that different approaches for cloud detection products from the 2 missions could lead to different results. See also specific comment below.

AC: We agree with the statement that different approaches for cloud detection products from the 2 missions could lead to different results. But, the idea of the paper was not to reconcile cloud product by "tweaking" the cloud detection algorithm, but to compare the fundamental differences. Therefore, here we used the same cloud detection for the two system. We agree that after having fully understood and quantify the differences due to the 2 systems (like we try to do here), the future work will include the algorithm adaptation to retrieve the same clouds and to build a long-term cloud record. We added the corresponding text in the conclusion as an interesting and exciting outlook.

**Specific comments**

RC: Page 1, line 22: "the ALADIN product demonstrates lower sensitivity because of lower backscatter at 355 nm": This statement is not clear. The backscatter at 355 nm is not expected to be lower than at 532nm. Please explain and revise accordingly.

AC: This is an important comment made by all three reviewers. Indeed, there was a confusing explanation regarding the particulate backscatter and we apologize for this. As we wrote in response to the Reviewer #1's question, we meant the contribution of the particles to the total (particulate + molecular) signal. Even though the total backscatter is larger at 355nm, the particulate part can be buried in molecular return because the molecular backscatter is larger at 355nm while the backscatter from cloud particles is about the same. If the signal-to-noise ratio is small, then the cross-talk correction will be noisy and the particulate signal will be retrieved with large uncertainty.

RC: Page 2, line 43: "Despite an excellent daily coverage and daytime/nighttime observation capability (Menzel et al., 2016; Stubenrauch et al., 2017), the height uncertainty of the cloud products retrieved from the observations performed by these spaceborne instruments is limited by the width of their channels' contribution functions, which is on the order of hundreds of meters, and the vertical profile of the cloud cannot be retrieved with accuracy needed for climate feedback analysis." The sentence is confusing. Consider revising to make it easier to follow. Possible suggestion: "…is limited by the width of their channels' contribution functions (which is on the order of hundreds of meters), and their uncapability to retrieve the vertical profile of the cloud with accuracy needed for climate feedback analysis.

AC: Thank you for this suggestion, we have simplified the text of this paragraph.

RC: Page 2, line 47: "This drawback is eliminated by active sounders, the very nature of which is based on altitude-resolved detection of backscattered radiation, and the vertical profiles of the cloud parameters are available from the CALIOP (Cloud-Aerosol Lidar with Orthogonal Polarization) lidar (Winker et al., 2003)and CloudSat radar (Stephens et al., 2002) since 2006, CATS (Cloud-Aerosol Transport System) lidar on-board ISS provided measurements for over 33 months starting from the beginning of 2015(McGill et al., 2015).": Too big sentence, difficult to read. Consider revising.

AC: We have simplified it, thanks.

RC: Page 4, line 106: "In Fig.1(a-c), we show the observation geometry and sampling of ALADIN's L2A product as well as three variables retrieved from its observations..": consider revising as: "…as three simulated variables that can be retrieved from its observations..".

AC: Since other Reviewers found this plot difficult to understand, we have replaced it with a 3D view of the orbits and observation geometries. Correspondingly, the description of Fig. 1 has changed.

RC: Page 4, line 106: "In Fig.1(a-c), we show the observation geometry and sampling of ALADIN's L2A product as well as three variables retrieved from its observations..": consider revising as: "…as three simulated variables that can be retrieved from its observations..".

AC: Thank you for the suggestion, but in the new version of the manuscript we have a different Fig. 1 with a somewhat different discussion.

RC: Page 4, line 120: "The cloud variability along the satellite's track has been estimated from the gridded EAMv1 data using the parameterization of (Boutle et al., 2014). Figure1 also serves as an illustration to theoretically achievable cloud detection agreement discussed below.": Although the cloud variability is estimated, in the plot the scene is cloud free. As the paper mainly investigates clouds, it would be interesting to have a cloudy demonstration also in addition to Figure 1.

AC: Fig. 1 does not exist in its previous form anymore, but in any case, the scene was not cloud free. The horizontal structures with large ATB values corresponded to the clouds.

RC: Page 4, line 123: "…scattering ratio (SR)..": Please write how the scattering ratio is calculated.

AC: This is a good point. In the new version of the manuscript, we have a whole new section (Sect. 3) dedicated to the definitions and formalism.

RC: Page 4, line 124: "An important companion of such a column is a corresponding quality flag column,…… which can be then compared with that of CALIOP.": The description is vague, please write more clearly what filtering you used in the data.

AC: We have updated the text to "The important companions of these profiles are quality flag columns. For our analysis, we kept only the layers, which are marked either by a high Mie SNR flag or by high Rayleigh SNR flag, and by a flag indicating an absence of signal attenuation."

RC: Page 5, line 141: "Since the CALIOP is not a HSRL, the detailed information on AMB and APB is not available, and one has to compare the SR products.": One could also use the temperature and pressure profiles from NWP (provided with Aeolus & CALIPSO) to produce the particulate backscatter coefficient, and convert/compare these parameters. So this part should be revised to highlight the choice of this study and not state it as the only option.

AC: Thank you for this suggestion, that's exactly how it's done in the new version of the manuscript. There's a small correction, though – the molecular backscatter coefficient is recalculated using P/T profiles, and not the particulate one.

RC: Page 5, line 145-150: "The choice of the fitting parameter is not crucial for the purposes of the present work … collocated data.": I strongly advise the authors to follow the comment of the first reviewer regarding the wavelength conversions. Alternatively, if they decide to keep the analysis as is, then please provide a detailed discussion on the uncertainties induced from this simplified conversion.

AC: For the new version we have updated the wavelength conversions and we discuss the uncertainties associated with it.

RC: Page 6, line 167: "To avoid the risks associated with the solar contamination, we picked up only the night-time cases": As Aeolus is in dusk-dawn, still variability is expected in the PBL with the CALIPSO nighttime observations above land. Can you comment on that in the manuscript?

AC: This is a valid point and, indeed, the diurnal cycle can spoil the comparison. Our answer is in our Fig. 3 (now Fig. 4), which estimates the diurnal effects along with the geometric and sampling differences. In addition, we rebuilt our new Fig. 5 (SR-height histograms) and Fig. 7 (cloud fraction profile per latitude) for the daily data without temporal difference filtering (these versions are not shown in the manuscript). In this approach, the diurnal effects are compensated because both local times are used for both instruments. Still, the SR-height histograms (Fig. 5) and cloud fraction profiles (Fig. 7) plots look about the same for this enhanced dataset as they do for a subset used in the manuscript, so one can conclude that the diurnal effects cannot explain the observed behavior.

RC: Page 6, line 172: "…we have performed a numerical experiment using the same calculated data as we used in Fig.1": Shouldn't they be stated as "simulations"?

AC: This is correct, but now we have a different Fig. 1 and a new section dedicated to the simulations, so this phrase does not exist anymore.

RC: Page 6, line 173 – 180: "This time… the passive observations": It is very hard to follow the approach. A scheme/flowchart would be useful

AC: We added a flowchart and we simplified the text, thanks for the suggestion.

RC: Page 6, line 182: "Overall, we considered about 1E5 pairs of pseudo-collocated data and we present the results of cloud detection in Fig.3": Please include also the region and season(s) used to produce these pseudo-collocated data, which represent the outputs of Fig. 3.

AC: We have updated the text of the paragraph and added a flowchart (Fig. 3). Briefly, we used 15 simulated orbits of one day in autumn equinox that cover both hemispheres and give, therefore, a representative snapshot of various atmospheric scenarios.

RC: Page 6, line 184: "or each altitude bin, the cloud detection agreement is a ratio of a number of cases when both instruments have detected a cloud (SR>5) ….": Please elaborate this choice of cloud cut off (e.g. literature) and comment on the uncertainties on the cloud detection induced from this choice for different altitudes. Could you include in results (Figure 3) and discuss, the percentage of the clouds missed to be detected, from the 2 sensors in your simulation, with the presented methodology?

AC: As for the choice of cutoff, we'd like first to refer to our answers to Reviewer #1's questions and to the two definitions of SR existing in the community. Indeed, a threshold applied to the SR defined as in Eq. 2 of present version of the manuscript should be altitude-dependent. But, as it is shown in (Chepfer et al., 2008, 2013) a fixed threshold can be applied to a SR defined as in Eq. 3 of the manuscript to estimate the difference between the two lidars. Future work will include a more advanced cloud detection algorithm to build a long-term cloud record. But this will be a whole new study.

RC: Page 7, section 3.1. It should be stated clearly in the section that the discussion refers to the SR retrieved products used in this study from the 2 sensors. As for example, a study with the cloud statistics from the Atlid L2A and CALIPSO L2 backscatter coefficient product products may provide different results.

AC: This is true, we hope that the new title clarifies that point.

RC: Page 8, line 224: "In Appendix A, we demonstrate the correlation between individual pairs of CALIOP and ALADIN SR profiles; the conclusion of this exercise is that it justifies using Eq.1, but the uncertainties of the analysis do not allow to refine the conversion coefficients". This statement is very strong. One could refine the conversion coefficients, independently of the uncertainties of the analysis. I support that the authors should formulate this statement to correctly reflect the choices and limitations.

AC: In the new version of the manuscript, we do not use Eq. 1 and we do not want to retrieve or validate its parameters anymore, so we do not seek to rebuild this plot.

RC: Page 8, line 229: "This observation gives a hint that the instrumental part provides the backscatter information sufficient for some cloud detection up to 20km, but the detection algorithm suppresses noisy solutions." This sentence is not clear. Please improve the phrasing.

AC: We added some explanations after this sentence.

RC: Page 8, line 246: "Below, we will also discuss the YES_YES statistics normalized to cloud amount, but at this point we also want to study the other cases, which cannot be normalized this way" Consider to improve the phrasing.

AC: We have rewritten this section.

RC: Page 9, line 283: "This exercise is not aimed at revealing any altitude offset in backscatter signal registration, because this part of experimental setup is robust in both instruments". Consider improving the phrasing.

AC: We have changed it to "We note that we are not looking for an altitude offset here. The altitude detection of both instruments is beyond question. Instead, we would like to check …"

RC: Page 9, line 10: "For each local peak found, we have searched for a peak or for a maximal value of CALIOP's SR profile in the vicinity of ±3km from the peak height determined from ALADIN". Consider including the information that only the 82% of the clouds are used for this comparison (according to the statistics presented in line 296-297.

AC: We added the proposed information in the following form: "By imposing the ±3 km search criteria, we filter out about 12% of the cases linked to natural variability, but at the same time we lower the rate of picking up the peak from a different cloud layer."

RC: Page 9, line 304: "As for the clouds between ~3km and ~10km height, the height sensitivity effects skew the effective cloud height detected by ALADIN downwards by 0.5−1.0km", It is not clear which are the high sensitivity effects between 3 to 10 km. Maybe the authors could summarize them in a sentence again here. Also, please comment to what extent could the actual 100-km-cloud-variability at these altitudes be responsible for the deviation in the altitudes seen by Aladin and Caliop in these altitudes. It is not clear if the authors point out on the Aeolus capability to detect the top of the cloud, on the SR methodology capability for the same, or on the effect of the natural variability between the 2 instruments on their products.

AC: We have updated the figure due to an improved recalculation of SR. The text has been updated, correspondingly. As for the possibility of 100km variability to be responsible for the observed shift, it is unlikely. The very nature of this variability is random and we do not expect it to have a bias. Moreover, the figure does not change that much if we loosen the collocation criteria, thus adding even more random variability.

RC: Figure 1: "…ALADIN's observation paths for centers of averaged profiles …": How they are averaged? In Aladin L2A resolution?

AC: We have a new version of Fig. 1 and the caption is now different, too.

RC: Figure 1: " This inclination is schematically shown as an inclined line lying in lidar curtain plane whereas the real projection to the same plane should be a vertical line": This part is hard to understand. Same comment for the part inside the manuscript.

AC: This figure has been replaced with a 3D view and the text has been modified correspondingly.

RC: Figure 2: Can the authors comment on the absence of collocated points between 0-60° lon at Δtime < 6hrs?

AC: This is a good point. The problem is purely technical: in this part, the data at 6 h difference come from another day and our collocation used the same day files. The collocation procedure is already heavy enough on resources, so we opted out of reading the other day's files. Technically, this is possible, but practically we would get only ~10% more of the collocated cases in the geographic area, which is not crucial for the comparison.

RC: Figure 7: No data is difficult to be distinguished from the -2km color, both have dark purple. Consider changing the no data color.

AC: We have changed the no data color.

RC: Figure 9: Consider adding the colorbar here also in the upper panel. Additionally, consider stating what the error bars account for.

AC: We have merged old Fig. 8 and Fig. 9 to a new Fig. 10. Correspondingly, all color panels share now the same color bar. As for the error bars, they correspond to r.m.s. of 1-week chunks of analyzed altitude subsets.

RC: Figure A1: The red points are not scaled in the same frequency ranges as the occurrence frequencies. Wouldn't that be better?

AC: This figure was removed from the new version of the manuscript.

**Technical corrections**

RC: Page 4, line 101: "According to Flamant et al. (2017)."

AC: Fixed.

RC: Page 6, line 182: "Ansmann et al. (2007)"

AC: We do not quote this work in this context anymore. Please, see the next-to-last answer to the Reviewer #1 comments.

RC: Page 7, line 195: "…between the two products.."

AC: This sentence has been rewritten

RC: Page 7, line 200: "..for the thw instruments"

AC: Fixed

RC: Page 7, line 203: "Analyzing the Fig. 4"

AC: Fixed

RC: Page 8, line 242: consider rephrasing to "from the sensitivity study.."

AC: This part has been rewritten

RC: Page 8, line 237: consider rephrasing to "..behavior of the SR cloud detection product agreement"

AC: We have updated the phrasing here.

RC: Figure 3: "...to the total number of simulations .."

AC: The whole caption of Fig. 3 (now Fig. 4) is different in the new version

RC: Figure 7: "...+-3km vertical vicinity…

AC: Fixed, thanks.

---

## Author Response (AR2)

We thank the Reviewer for his/her new round of comments on the paper. The responses to general and specific comments are given below. We marked the reviewer's and the author's comments by "RC:" and "AC:", respectively.

**General comments**

RC: In my opinion it would be very useful to discuss the coarse Aeolus resolution (vertical and horizontal) compared to the raw Calipso resolution based on the numerical simulation. This would give a hint how much information is lost due to the resolution matter and would give valuable advise for the future planned space lidar missions. It should be also easily achievable with the tools already developed.

AC: We agree that this is a very important issue, which has been addressed in the previous works (e.g. Chepfer et al., 2013) and is discussed in application to Aeolus (see e.g. the recent presentations on Aeolus L2A processors by Dabas et al., and Donovan et al., 2021 at the 3rd Aeolus L2A working meeting, 16.12.2021). We have performed the requested simulations and put them in a new Sect. 4.2. The advice for the future lidar missions can be formulated similar to Nyquist criterion for signals. In application to cloud observations, this means that the sampling rate converted to meters along the track should be smaller than or comparable to the typical size of the smallest optically thick clouds we want to resolve, that are typically shallow liquid clouds in the boundary layer. So, we speak about the resolution of about 100-300m, but, as we show in new Section 4.2, even going down to 3 km from 87 km fixes a lot of problems. We put a stress on thick clouds because of the following effect: if a scene containing optically thick clouds and clear sky is averaged, strong signal from clouds will trigger the cloud detection and the whole averaged area will be marked as cloudy thus leading to an overestimate of cloud amount. On the other hand, the downside of increasing the resolution is also obvious – in the case of an optically thin cloud the weak signal-to-noise ratio will prevent its detection. Correspondingly, it is highly advised to use a high resolution, which is just enough to tell the thick cloud from clear sky and perform additional averaging, if needed, to detect thinner clouds (Chepfer et al., 2013). We addressed these issues specifically for ALADIN in the new Sect. 4.2, even if it is not fundamentally new information.

RC: It is not clear for me if you adapted the Calipso grid also to the Aeolus vertical resolution. As you know, the vertical resolution of Aeolus is changing with geographical region and also special range-bin settings are chosen for specific time periods. Have you considered this? If yes, please state this explicitly, if not please do so.

AC: We chose not to use variable vertical grid of ALADIN. Instead, we fixed a 1 km vertical grid everywhere and filled it with the corresponding CALIPSO and ALADIN data. We added this information to Section 3.3 and Section 3.4. We'd like to note that the sensitivity of the cloud detection to the vertical resolution has been discussed and quantified in (Chepfer et al. 2010) using CALIPSO raw data.

RC: With respect to the comment above I wonder how useful the section 5.4 (Cloud altitude detection sensitivity) is. Considering 1-2 km thick range bins of Aeolus in the upper atmosphere, I do not understand the usefulness of your chosen approach and how Fig 9. can be created with this high resolution. In my opinion, section 5.4 is only useful, if you use Caliop raw resolution data and discuss your findings with respect to the coarse vertical resolution of Aeolus. Then, there is also no need to allow the large vertical range of +-3 km but limit it to 2 neighbouring range bins of Aeolus in the vicinity of a cloud. And a final conclusion would be good: How superior or not is Calipso with respect to thin cloud detection based on its raw resolution? Or in other words: Is the current resolution sufficient or not for cloud detection and what resolution would you recommend.

AC: As for the resolution of Fig. 9, one has to keep in mind that this is an average distribution and that each bin corresponds to more than one colocation. In this case, if, for example, the ALADIN cloud was found 10 times at -2 km, 27 times at -1 km and 63 times at 0 km with respect to CALIOP peak, the mean downward shift would be equal to 0.47 km. As for the last question, we address it in new Section 4.2 along with the horizontal resolution.

RC: And for what is a mean cloud height useful?

AC: If the mean is composed of high- and low-level clouds mixed together then this information is of poor use. Nevertheless, the mean cloud height might be useful for future use of ALADIN data for climate-oriented studies. Indeed, mean cloud heights are used in several studies coupling CALIPSO data with radiation to characterize co-variabilities and better understand the climate system. In particular, it is used in cloud feedback analysis (eg. Vaillant de Guélis et al. 2017, 2018). Other examples of studies using CALIPSO cloud mean height and radiation are presented in (Guzman et al., 2017; Frey et al., 2018; Morrison et al., 2019; Perpina et al., 2021).

RC: In my opinion Sec. 5.5 can be omitted, it could be covered by one statement: No temporal variation was found during for the analysed period. The plots confirming this could be put in the supplementary material. But in my opinion, it distracts the focus of the current paper by opening a new side topic which is not helpful for the paper structure.

AC: We see the point, but the whole study stems from the Cal/Val activity of ALADIN, for which the temporal stability of the instrument and its products was the key issue. The instrument team had to fix the laser induced contamination, optical losses, and hot pixel issues and it would be unfair to hide this under one sentence. With this section, we wanted to highlight that the aforementioned issues were fixed at least down to uncertainties of our analysis that it a good result.

**Specific comments:**

RC: Line 25: Improve phrasing

75  AC: we simplified the second part to "considering the differences between the instruments".

RC: Bley et al., 2021: Is referring to confluence which is not publicly available. Thus, it cannot be referenced. Please find another source of reference or provide the information on a public repository or in your supplementary material.

AC: We agree that the referenced material is unavailable for the general public, but we cannot copy-
80  paste the confluence pages, so we changed the reference to "ESA, ESA News, ALADIN overview and timeline of the RBS settings, available at https://earth.esa.int/eogateway/instruments/aladin/overview-of-the-main-wind-rbs-changes, 2021." and we ask an advice of the AMT technical team whether such a reference is acceptable.

RC: Line 176: CALIOP also emits 1064 nm.

85  AC: Fixed, thanks.

RC: Line 185: The multiple scattering coefficient is used for CALIOP. Did you also consider multiple scattering effects for Aeolus?

AC: The ALADIN's field of view is much narrower than that of CALIOP and multiple scattering effects are usually not considered for this instrument. Nevertheless, (Donovan et al., 2020) estimate
90  them to be non-negligible, so we took a value of 0.9 for our recalculation procedure. We also tested the value of 0.7 and found that it affected only the low-level clouds retrieved from recalculated SR_532 for ALADIN. We added a short discussion of this value after new Eq. 8.

RC: Line 242: Writing down the final formula would be really helpful here.

AC: We added a final formula.

95  RC: Eq. 9: Must be wrong. You subtract the same parameter from each other.

AC: Fixed, thanks.

RC: Line 255: Eq. 10: How useful is the term cloud fraction based on 87 km resolution?

AC: Indeed, it is not useful for a single ALADIN bin, but the equation holds true for zonal averages and other types of averages.

100  RC: Line 287: What is Aladins original laser pulse frequency rate?

AC: It is equal to 50Hz, we added this information to this line.

RC: Caption Fig. 4: Please use once again full words for your parameters: E.g.: CDA (cloud detection agreement) or cloud fraction CF...and so on. So that the reader can understand the figures without digging in the text.

AC: We have updated the figure caption, thanks.

RC: Line 305: would be great to have the cloud fraction of Caliop at native resolution for comparison. This could give an indication evidence how much of clouds are missed due to the coarse horizontal resolution of Aeolus.

AC: We address this problem in new section 4.2 (see also the answer to the first general comment). The coarse resolution of ALADIN might lead both to under- and overestimation of cloud cover for certain type of scenes such as boundary layer shallow liquid in the tropical subsidence boundary layer that represents 2/3 of the surface of the entire Tropical belt (30S/30N).

RC: Line 317: would be nice to write these latitude bands directly in figure 5.

AC: We added the latitude band limits on top of each column

RC: Line 327: Phrasing: "Both rows show….."

AC: we have rephrased it to "zonal averages agree reasonably well up to ~3km altitude;"

RC: Line 335: How can a peak be detected within 1 km when the vertical resolution is 1 km?

AC: Please, see our explanations in "General comments" section.

RC: Line 338: I guess Fig 5 is meant.

AC: Fixed, thanks.

RC: Line 343: "This gives a hint that the instrumental part itself provides the backscatter information sufficient for cloud detection up to 20 km, but the detection algorithm suppresses noisy solutions. "◊ I cannot follow this conclusion. Can you give more detail?

AC: Unfortunately, we cannot provide more details as we do not work with L2A algorithms. According to what we've heard at the Aeolus Workshops and to the common knowledge of the retrieval algorithms, it is quite natural that the unregularized solution will be rejected if some parameter the r.m.s. of the difference between the measured signal and the simulated one is large or if the layer-per-layer retrieval does not converge in the case of a noisy signal. The idea of this paragraph was to tell the reader that the instrument itself can measure the signal at these heights, but the signal quality is not always enough to obtain a good solution.

RC: Line 351: Fig 5 or 6 f?

AC: We meant Fig. 5f here since it shows an averaged plot with a noticeable PSC signature. Please, see the next comment regarding the overestimation of SRs for PSCs in our approach.

RC: Line 351: "…confirms this assumption because the vertical extent and the composition of these clouds yield a stronger SR signal than that for the cirrus clouds (Noël et al., 2008)." again I cannot follow that conclusion. Can you explain more why the PSC detection confirms your theory with respect to the lower SNR of Aeolus?

AC: The observed effect is likely explained by an artificial increase of SR in PSC area due to our conversion procedure of ALADIN data to SR'532. This conversion assumes the particulate extinction and backscatter are the same at 532 and 355 that is not true for PSCs composed of STS droplets (Jumelet et al. 2009). At the same time, the signal itself should be strong enough to be detectable, so the initial logic behind the aforementioned lines is also true. This is now discussed in Sect. 5.1.

RC: Line 355: Fig 6 instead of 5?

AC: Fixed, thanks. We note that the updated version of the manuscript uses new enumeration due to a new figure added after Fig. 3.

RC: Line 376: I cannot see this 40% according to the given color bar. At which plot shall I look at?

AC: 40% is a ratio between the values seen in Fig. 8a (~6%) and those estimated in Fig. 4 (blue dashed curve, theoretical CDA with noise), which is about 10% at heights above ~8 km. One can also divide the number of YES_YES cases by a sum of YES_YES and YES_NO cases to get the same 40% for high clouds.

RC: Line 380: phrasing to be improved:"… false detection rate of both instruments is low that is a good result."

AC: We have rephrased this sentence to "This indicates that the false cloud detection induced by a small SNR in cloud-free area is rare for both instruments. We consider this to be a good sign as it shows the stability of the ALADIN retrieval algorithm for weak signals."

RC: Line 402: "To explain the observed behavior, one needs to have either smaller $\alpha part(\lambda,z)$ values, or larger $\beta part(\lambda,z)$ values, or both. " I cannot follow this conclusion. Can you please explain.

AC: Here, we hypothesize that not all variables are always accurately retrieved. This allows us to discuss what bias of which parameter would explain the observed discrepancy. We are focusing on an unexpected retrieval of a peak beneath a thick cloud and we speculate that for these cases either the extinction coefficient at and above the considered point is unrealistically low, or the retrieved backscatter coefficient is unrealistically high, or both.

RC: Line 405: Please improve phrasing: "The uncertainties of the parameters used for their estimate are small (Bucholz, 1995; Ciddor, 1996), so they cannot give preference to the low-level clouds and suppress the higher ones."

AC: We removed the part after the references because it is redundant.

RC: complete section 5.3 should be improved concerning language.

AC: We have simplified this section and improved the phrasing.

RC: Line 411: "To reduce the low-level clouds in Eq. 1, " why reduce low level clouds? I thought no_Yes, means no Caliop cloud but an Aeolus cloud. this does not make sense to me.

AC: This is true, NO_YES means that CALIOP did not detect a cloud and ALADIN saw it. The problem is that in the discussed case ALADIN was not supposed to detect any cloud because there was a thick cloud above whereas the ALADIN's wavelength and observation geometry should lead to full attenuation of the signal at a higher altitude for ALADIN than for CALIOP. Therefore, we considered the ways of reducing the clouds in such situations. In the updated version of the manuscript, we have added the values of multiple scattering coefficient to the discussion in (4c) and we have rewritten this paragraph.

RC: Line 437: Fig. 9 instead of 8?

AC: Yes, thanks.

RC: Line 464: I think, it is the first time that the period which was analysed is mentioned. Please do so earlier.

AC: In the version under discussion, the analyzed period was mentioned in the Abstract (line 18) and then in the text (lines 121 and 464).

RC: Line 486: phrasing: "cloud changes" what are cloud changes?

AC: We have rephrased the sentence: "Performing climate studies and building a long-term cloud record with the help of these instruments requires understanding…"

RC: Caption Fig 2: <1h should be deleted right?

AC: Fixed, thanks.

**References used in this text**

Chepfer H., S. Bony, D. Winker, G. Cesana, JL. Dufresne, P. Minnis, C. J. Stubenrauch, S. Zeng, 2010: The GCM Oriented Calipso Cloud Product (CALIPSO-GOCCP), J. Geophys. Res., 115, D00H16, doi:10.1029/2009JD012251.

Chepfer H., G. Cesana, D. Winker, B. Getzewich, and M. Vaughan, 2013: Comparison of two different cloud climatologies derived from CALIOP Level 1 observations: the CALIPSO-ST and the CALIPSO-GOCCP, J. Atmos. Ocean. Tech., doi.10.1175/JTECH-D-12-00057.1

Frey, W.R., Morrison, A.L., Kay, J. E, Guzman R., and H. Chepfer, 2018: The combined influence of observed Southern Ocean clouds and sea ice on top-of-atmosphere albedo, J. Geophys. Res. Atmos., DOI: 10.1029/2018JD028505

Guzman, R., Chepfer, H., Noel, V., Vaillant de Guelis, T., Kay, J.E., Raberanto, P., Cesana, G., Vaughan, M. A., and D. M. Winker 2017: Direct atmosphere opacity observations from CALIPSO provide new constraints on cloud-radiation interactions, J. Geophys. Res. Atmos., DOI: 10.1002/2016JD025946

Morrison, A. L., Kay, J. E., Frey, W. R., Chepfer, H. and R. Guzman, 2019: Cloud response to Arctic Sea Ice Loss and Implications for Future Feedbacks in the CESM Climate Model, J. Geophys. Res. Atmos., DOI: 10.1029/2018JD029142

Perpina, M., Noel, V., Chepfer, H., Guzman, R., Feofilov, A.G., (2021) Link Between Opaque Cloud Properties and Atmospheric Dynamics in Observations and Simulations of Current Climate in the Tropics, and Impact on Future Predictions, https://doi.org/10.1029/2020JD033899, J. Geophys. Res. Atmos

Vaillant de Guélis T., H. Chepfer, Noel, V., Guzman, R., Bonazzola, M., and Winker, D. M, 2018: Space lidar observations constrain longwave cloud feedback, Nature Scientific Reports, 8:16570 | DOI:10.1038/s41598-018-34943-1.

Vaillant de Guélis T., H. Chepfer , V. Noel, R. Guzman, P. Dubuisson, D. M. Winker, and S. Kato, 2017: Link between the Outgoing Longwave Radiation and the altitude where the space-borne lidar beam is fully attenuated, Atmos. Meas. Tech., doi:10.5194/amt-2017-115